# KEAP1 retention in phase-separated p62 bodies drives liver damage under autophagy-deficient conditions

Shuhei Takada [1], Nozomi Shinomiya[1], Gaoxin Mao [1], Hikaru Tsuchiya [1,2], Tomoaki Koga [3], Satoko Komatsu-Hirota[1], Yu-shin Sou[4], Manabu Abe [5], Elena Ryzhii[6], Michitaka Suzuki [6], Mitsuyoshi Nakao [3], Satoshi Waguri [6], Hideaki Morishita[1,7] & Masaaki Komatsu [1,2 ✉]

## Abstract

Phase-separated p62 bodies activate NRF2, a key transcription factor for antioxidant response, by sequestering KEAP1, which targets NRF2 for degradation. Although p62 bodies containing KEAP1 are degraded by autophagy, they accumulate in various liver disorders. Their precise disease role remains unclear. We show that excessive KEAP1 retention in p62 bodies and NRF2 activation are major causes of liver damage when autophagy is impaired. In mice with weakened or blocked p62-KEAP1 interactions, KEAP1 retention and NRF2 activation under autophagy-deficient conditions were suppressed. Transcriptome and proteome analyses reveal that p62 mutants unable to bind KEAP1 normalize the expression of NRF2 targets induced by defective autophagy. Autophagy deficiency causes organelle accumulation, especially of the ER, regardless of p62 mutation. Liver damage and hepatomegaly resulting from autophagy suppression markedly improved in mice carrying p62 mutants, particularly those with blocked KEAP1 binding. These findings highlight excessive KEAP1 retention in p62 bodies and defective organelle turnover as key drivers of liver pathology, underscoring the significance of phase separation in vivo.

**Keywords** p62; KEAP1; NRF2; Liquid–Liquid Phase Separation; Stress Response
**Subject Categories** Autophagy & Cell Death; Molecular Biology of Disease; Organelles

## Introduction

In cells, phase-separated droplets formed through ubiquitin chains serve as sites for protein degradation. Notable examples of these droplets include proteasome foci and p62 bodies. Proteasome foci rapidly form in the nucleus in response to high osmotic stress. Their formation is triggered by phase separation through multi-valent interactions between polyubiquitinated proteins and the shuttle molecule RAD23B, with proteasomes and the ubiquitin-selective chaperone p97/VCP accumulating at these sites. Hyperosmotic stress induces nucleolar stress, resulting in orphan ribosomal proteins that are excluded from ribosome complexes and leak from the nucleolus into the nucleoplasm. Proteasome foci participate in degrading these ribosomal subunits (Yasuda et al, 2020). p62 bodies, on the other hand, are liquid droplets that form in the cytoplasm through multivalent interactions between poly-ubiquitinated proteins and the ubiquitin-binding protein p62. These p62 bodies, along with their client proteins (proteins contained within p62 bodies), are subsequently degraded by autophagy.

When proteins are denatured by oxidative stress, translation inhibition, or the presence of abnormal nascent peptide chains, they undergo ubiquitination. These ubiquitinated proteins then bind to p62, triggering liquid–liquid phase separation (Sun et al, 2018). Normally, p62 forms oligomers through its self-oligomerization Phox1 and Bem1p (PB1) domain, resulting in fibrillar p62 structures (preprint: Berkamp et al, 2024; Ciuffa et al, 2015; Jakobi et al, 2020). These p62 fibrils bind to ubiquitin chains, promoting liquid–liquid phase separation and the formation of liquid-like p62 bodies (Sun et al, 2018; Zaffagnini et al, 2018). When the Serine 405 residue in the ubiquitin-associated (UBA) domain of p62 is phosphorylated by Unc-51-like kinase 1 (ULK1), or the Serine 403 residue is phosphorylated by TANK-binding kinase 1 (TBK1) or Casein kinase 2 (CK2), p62's binding affinity to ubiquitin chains is enhanced (Matsumoto et al, 2011; Pilli et al, 2012). The p62 body then matures by incorporating client proteins such as Kelch-like ECH-associated protein 1 (KEAP1) (Ikeda et al, 2023), Vault particles (Kurusu et al, 2023), and selective autophagy receptors like Next to BRCA1 gene 1 (NBR1) and Tax1-binding protein 1 (TAX1BP1) (Turco et al, 2021). The formation of an isolation membrane on the p62 body occurs when FAK family

[1]Department of Physiology, Juntendo University Graduate School of Medicine, Bunkyo-ku, Tokyo, Japan. [2]Autophagy Research Center, Juntendo University Graduate School of Medicine, Bunkyo-ku, Tokyo, Japan. [3]Department of Medical Cell Biology, Institute of Molecular Embryology and Genetics, Kumamoto University, Kumamoto, Japan. [4]Department of Cell Biology and Neuroscience, Juntendo University Graduate School of Medicine, Bunkyo-ku, Tokyo, Japan. [5]Department of Animal Model Development, Brain Research Institute, Niigata University, Chuo-ku, Niigata 951-8510, Japan. [6]Department of Anatomy and Histology, Fukushima Medical University School of Medicine, Fukushima, Japan. [7]Department of Molecular Cell Biology, Graduate School of Medical Sciences, Kyushu University, Higashi-ku, Fukuoka, Japan. ✉E-mail: mkomatsu@juntendo.ac.jp

kinase-interacting protein of 200 kDa (FIP200), the most upstream factor in autophagosome formation, binds directly to p62 or to TAX1BP1 (Turco et al, 2019; Vargas et al, 2019). This membrane formation along p62 bodies continues as ATG8 family proteins localized on the isolation membrane bind to p62, NBR1, and TAX1BP1, and as wetting effects between the isolation membrane and p62 bodies reinforce the process (Agudo-Canalejo et al, 2021; Kageyama et al, 2021). Ultimately, p62 bodies containing a high concentration of ubiquitinated proteins are degraded via autophagy. Thus, p62 and its phase separation facilitate the efficient degradation of ubiquitinated proteins, thereby contributing to proteostasis.

Like many other cellular droplets, p62 bodies are not merely passive entities degraded by autophagy; they also have active functions. A key physiological function of p62 bodies is regulating the KEAP1-NF-E2 p45-related factor 2 (NRF2) pathway, a major oxidative stress response mechanism. NRF2 is a master transcription factor that induces the expression of genes encoding antioxidant proteins and detoxification enzymes. Under non-stress conditions, NRF2 binds to KEAP1, a substrate recognition adaptor of the Cullin 3 (CUL3)-type ubiquitin ligase, and is ubiquitinated and degraded by the 26S proteasome. When cells experience oxidative stress or electrophilic substances, specific cysteine residues on KEAP1 undergo oxidative modification, leading to the dissociation of KEAP1 from NRF2 or, alternatively, disruption of the KEAP1–CUL3 interaction. This stabilizes NRF2, allowing it to translocate to the nucleus and upregulate a series of NRF2 target genes (Yamamoto et al, 2018). In addition to this canonical pathway, p62 competitively inhibits KEAP1-NRF2 binding, activating NRF2 even in the absence of KEAP1 modification (Jain et al, 2010; Komatsu et al, 2010; Lau et al, 2010). This p62-mediated NRF2 activation is regulated by phase-separated p62 bodies. Normally, KEAP1 maintains equilibrium between the cytoplasm and p62 bodies. When the Serine 349 residue (Ser349) of p62 in phase-separated p62 bodies is phosphorylated by ULK1, the binding affinity between KEAP1 and p62 increases, leading to KEAP1 retention within the p62 body (Ichimura et al, 2013; Ikeda et al, 2023). Consequently, KEAP1-mediated ubiquitination of NRF2 in the cytoplasm is inhibited, resulting in NRF2 activation. Because the *Sequestosome 1* (*SQSTM1*) gene, encoding p62, is itself a target of NRF2 (Jain et al, 2010), it is thought that p62 bodies form in response to oxidative stress and that NRF2 activation is a logical outcome. In addition to *SQSTM1*, NRF2 targets include genes encoding proteasome subunits (Kwak et al, 2003) and autophagy-related proteins (Pajares et al, 2016). Thus, the stress response mechanism mediated by p62 bodies is thought to be regulated by a balance of gene expression, protein degradation, and post-translational modification of p62 bodies. Previous studies have reported impaired autophagic degradation of p62 bodies in multiple mouse models of non-alcoholic steatohepatitis (NASH)-derived hepatocellular carcinoma (HCC) (Inokuchi-Shimizu et al, 2014; Nakagawa et al, 2014; Umemura et al, 2016; Zhang et al, 2015). Structures known as Mallory-Denk bodies, which are positive for p62, KEAP1, and ubiquitin, have been observed in various liver disorders, including NASH and HCC (Ichimura et al, 2013; Inami et al, 2011; Stumptner et al, 2002). These findings suggest that abnormalities in the turnover and/or function of p62 bodies are involved in the development of human liver diseases.

In this study, we generated *p62* knock-in mice with reduced KEAP1 retention (*p62^{S351A/S351A}*) or lacking KEAP1 (*p62^{T352A/T352A}*) in p62 bodies and crossbred them with *Atg7^{flox/flox}*;Alb-Cre mice, which exhibit severe liver damage, hepatomegaly, and abnormal activation of the NRF2 pathway mediated by p62 bodies (Komatsu et al, 2010). Due to *Atg7* deletion, both wild-type and mutant p62 accumulated, and p62 body formation was confirmed. In wild-type p62 bodies, high phosphorylation of p62 at Ser351 (equivalent to human Ser349) retained KEAP1, leading to NRF2 activation. In contrast, *p62^{S351A/S351A}* and *p62^{T352A/T352A}* bodies showed significantly reduced KEAP1 incorporation and suppressed NRF2 activation. Transcriptome and proteome analyses revealed NRF2 targets as a group prominently upregulated due to autophagy deficiency, which were normalized when KEAP1 was excluded from p62 bodies in mutant mice. Autophagy deficiency also caused organelle accumulation, particularly of the endoplasmic reticulum, independently of p62 mutations. The severe hepatomegaly and liver damage in *Atg7^{flox/flox}*;Alb-Cre mice were ameliorated as KEAP1 sequestration in p62 bodies decreased. These findings suggest that persistent activation of the p62 body-mediated stress response, combined with defective autophagic organelle turnover, contributes to the pathogenesis of liver disorder.

## Results

### Loss of *Atg7* in mouse livers leads to the accumulation of p62 bodies

First, to determine whether the p62-positive structures observed in *Atg7*-deficient hepatocytes (Komatsu et al, 2007) are bona fide p62 bodies—liquid droplets rather than solid protein aggregates—we isolated primary hepatocytes from 5-week-old *Atg7^{flox/flox}* and *Atg7^{flox/flox}*;Alb-Cre mice. Immunofluorescence analysis using a p62 antibody revealed a significant accumulation of round p62-positive structures upon *Atg7* ablation (Fig. EV1A). Electron microscopy further identified round fibrillar assemblies, characteristic of p62 bodies (preprint: Berkamp et al, 2024; Jakobi et al, 2020), along with aberrant organelles, accumulating in *Atg7*-deficient hepatocytes (Fig. EV1B). If these structures are indeed p62 bodies, they should be degraded by autophagy. To test this hypothesis, we introduced either wild-type ATG7 or an active-site mutant ATG7^{C572S} into *Atg7*-knockout hepatocytes using an adenoviral system. Expression of wild-type ATG7, but not the mutant, restored autophagy, as evidenced by LC3-I to LC3-II conversion (Kabeya et al, 2000) (Fig. EV1C). Furthermore, p62 protein levels and its phosphorylated form (p-S351)—hallmark of p62 bodies (Ikeda et al, 2023)—progressively decreased over time upon wild-type ATG7 expression, while remaining unchanged in ATG7^{C572S}-expressing cells (Fig. EV1C). Likewise, the expression of wild-type ATG7, but not ATG7^{C572S} significantly reduced KEAP1, a client protein of p62 bodies (Fig. EV1C). Immunofluorescence analysis further confirmed that the number and the size of p62-positive structures decreased following wild-type ATG7 expression, whereas ATG7^{C572S} had no effect (Fig. EV1D). Taken together, these findings demonstrate that the p62-positive structures accumulating in *Atg7*-deficient hepatocytes are bona fide autophagy-degradable p62 bodies.

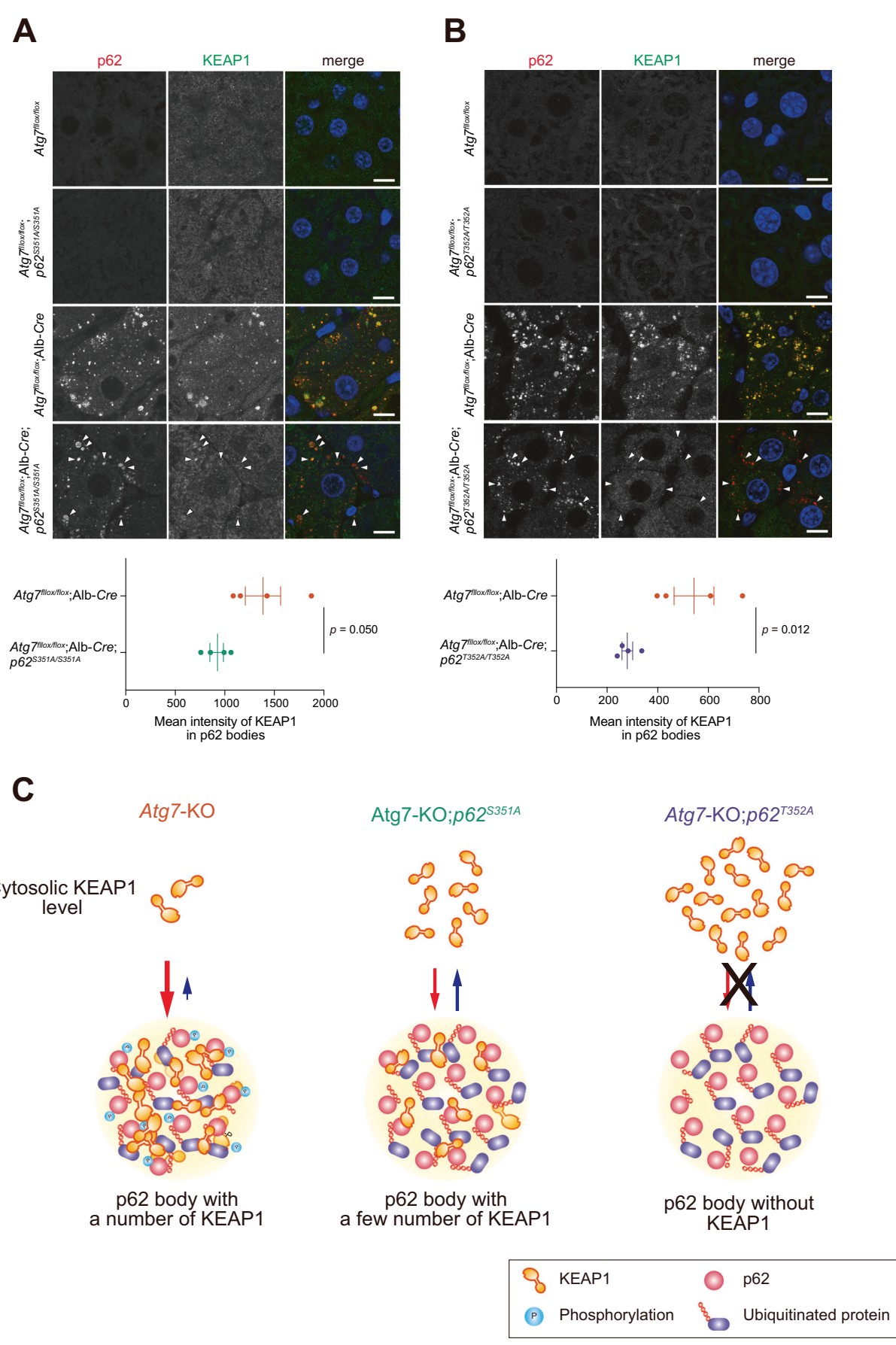

◄

**Figure 1.   KEAP1 dynamics in *Atg7*-deficient hepatocytes with different *p62* mutations.**

(A, B) Immunohistofluorescence analysis. Liver sections from *Atg7^flox/flox^*, *Atg7^flox/flox^;p62^S351A/S351A^*, *Atg7^flox/flox^;Alb-Cre*, and *Atg7^flox/flox^;Alb-Cre;p62^S351A/S351A^* mice (A) and from *Atg7^flox/flox^*, *Atg7^flox/flox^;p62^T352A/T352A^*, *Atg7^flox/flox^;Alb-Cre*, and *Atg7^flox/flox^;Alb-Cre;p62^T352A/T352A^* mice (B) at 3 month-old were immunostained with anti-p62 (left panels) and anti-KEAP1 (middle panels) antibodies. The right panels show the merged images of p62 (red) and KEAP1 (green). Arrowheads indicate examples of p62 bodies. Bars: 10 μm. The graph shows the mean intensity of KEAP1 in p62 bodies in the livers of each genotype (*n* = 4) mice. Vertical bars are means ± s.e. Statistical analysis was performed by Welch's *t* test. (C) Schematic diagram of KEAP1 retention within p62 bodies in three mouse lines: *Atg7^flox/flox^;Alb-Cre* (*Atg7*-KO), *Atg7^flox/flox^;Alb-Cre;p62^S351A/S351A^* (*Atg7*-KO;*p62^S351A^*) and *Atg7^flox/flox^;Alb-Cre;p62^T352A/T352A^* (*Atg7*-KO;*p62^T352A^*) mice. Source data are available online for this figure.

## KEAP1 is sequestered into p62 bodies based on its binding affinity to p62 in vivo

To investigate the physiological significance of KEAP1 sequestration in p62 bodies within the liver, we developed *p62* knock-in mice with diminished KEAP1 binding (*p62^S351A/S351A^*) or entirely lacking KEAP1 retention (*p62^T352A/T352A^*), in p62 bodies (Komatsu et al, 2010). These knock-in mice were then crossbred with *Atg7^flox/flox^*;Alb-Cre mice, which exhibit severe liver damage, hepatomegaly, and dysregulated activation of the NRF2 pathway mediated by p62 bodies (Komatsu et al, 2010). This approach allowed us to assess the role of altered KEAP1-p62 interactions in the context of liver pathology. As shown in Fig. 1A,B, double immunofluorescence analysis with p62 and KEAP1 antibodies revealed that p62 bodies formed in the hepatocytes of *Atg7^flox/flox^*;Alb-Cre;*p62^S351A/S351A^* and *Atg7^flox/flox^*;Alb-Cre;*p62^T352A/T352A^* mice, similar to those in *Atg7^flox/flox^*;Alb-Cre mice. However, both the size and number of p62 bodies in hepatocytes with the p62 mutant background tended to decrease compared to those with the wild-type p62 background (Fig. 1A,B). In autophagy-competent hepatocytes of *Atg7^flox/flox^*, *Atg7^flox/flox^*;*p62^S351A/S351A^*, and *Atg7^flox/flox^*;*p62^T352A/T352A^* mice, p62 bodies were rarely observed (Fig. 1A,B). The signal intensity of KEAP1 in p62 bodies in the hepatocytes of *Atg7^flox/flox^*;Alb-Cre mice was significantly higher than that of *Atg7^flox/flox^*;Alb-Cre;*p62^S351A/S351A^* or *Atg7^flox/flox^*;Alb-Cre;*p62^T352A/T352A^* mice (Fig. 1A,B). While KEAP1 was still retained in p62 bodies within *Atg7^flox/flox^*;Alb-Cre;*p62^S351A/S351A^* hepatocytes (Fig. 1A), it was nearly absent from p62 bodies in *Atg7^flox/flox^*;Alb-Cre;*p62^T352A/T352A^* hepatocytes (Fig. 1B). This approach enabled us to establish three distinct mouse lines: one accumulating p62 bodies that sequester KEAP1 (*Atg7^flox/flox^*;Alb-Cre: referred to as *Atg7*-KO), another with reduced KEAP1 retention in p62 bodies (*Atg7^flox/flox^*;Alb-Cre;*p62^S351A/S351A^*: referred to as *Atg7*-KO;*p62^S351A^*), and a third accumulating p62 bodies with minimal KEAP1 retention (*Atg7^flox/flox^*;Alb-Cre;*p62^T352A/T352A^*: referred to as *Atg7*-KO;*p62^T352A^*) (Fig. 1C). We also simply refer to *Atg7^flox/flox^* as cont., *Atg7^flox/flox^*; *p62^S351A/S351A^* as *p62^S351A^*, and *Atg7^flox/flox^*;*p62^T352A/T352A^* as *p62^T352A^*.

In the *Atg7*-KO background, ATG7 and LC3-II were nearly undetectable regardless of the presence or absence of the p62 mutation (Fig. 2A,B), indicating a blockade of autophagy. Ubiquitinated proteins significantly accumulated in the livers of *Atg7*-KO mice, and substitution of wild-type p62 with KEAP1-interaction-deficient mutants had little effect on this accumulation (Fig. 2A,B). Loss of *Atg7* resulted in increased levels of both KEAP1 protein and p62 protein, including the Ser351-phosphorylated form (Fig. 2A,B). While this increase in KEAP1 was reversed by replacing wild-type p62 with *p62^T352A^* (Fig. 2B), it persisted when replaced with *p62^S351A^* (Fig. 2A). Neither *p62^S351A^* nor *p62^T352A^* was recognized by the Ser351-phosphorylated p62 antibody. These results suggest that autophagy degrades KEAP1 incorporated into p62 bodies rather than cytosolic KEAP1. Indeed, KEAP1 protein

levels in *p62* knockout Huh-1 cells decreased when wild-type p62, p62^S349E^ (equivalent to mouse p62^S351E^), or p62^S349A^ (equivalent to mouse p62^S351A^) were expressed (Fig. EV2A). This decrease was not observed with p62^T350A^ (equivalent to mouse p62^T352A^) expression (Fig. EV2A). Ablation of FIP200 abolished the effects of wild-type p62, p62^S349E^ and p62^S349A^ (Fig. EV2B). Wild-type p62 and all p62 mutants formed p62 bodies; however, only those composed of the p62^T350A^ mutant failed to recruit KEAP1, irrespective of the cellular background—whether in *p62* knockout or *FIP200 p62* double knockout cells (Fig. EV2C,D). Thus, we concluded that KEAP1 translocation into p62 bodies is required for its autophagic degradation.

## NRF2 activation depends on the binding affinity of KEAP1 to p62 in vivo

As previously reported by us and others (Inami et al, 2011; Komatsu et al, 2010; Ni et al, 2014; Takamura et al, 2011), suppression of autophagy in mouse livers is associated with the induction of numerous NRF2 target genes. This induction has also been shown to be reversed by the simultaneous ablation of *Sqstm1*, which encodes the p62 protein, or *Nrf2*. To investigate whether KEAP1 retention in p62 bodies contributes to p62-dependent NRF2 activation, we performed mRNA-seq analysis on livers from the following mouse genotypes: cont., *Atg7*-KO, *p62^S351A^*, *Atg7*-KO;*p62^S351A^*, *p62^T352A^*, and *Atg7*-KO;*p62^T352A^*. As shown in Fig. 3A, the transcriptome profile of *Atg7*-KO;*p62^S351A^* was distinct from that of *Atg7*-KO and relatively similar to that of cont. In contrast, the transcriptome profile of *Atg7*-KO;*p62^T352A^* was more similar to that of cont. (Fig. 3B). We next analyzed differentially expressed genes (DEGs) among cont., *Atg7*-KO, and *Atg7*-KO;*p62^S351A^*. DEGs in Group 1 and Group 2 showed drastic changes due to *Atg7*-deficiency and a partial recovery with the p62^S351A^ mutation (Fig. 3C). Group 1 genes were associated with amino acid metabolism, lipid metabolism, and oxidation. Group 2 genes were linked to the NRF2 pathway and glutathione metabolism (Fig. 3D). Similarly, we analyzed DEGs among cont., *Atg7*-KO, and *Atg7*-KO;*p62^T352A^*. DEGs in Group 4 and Group 6 showed drastic changes due to *Atg7*-deficiency and a complete recovery with the p62^T352A^ mutation (Fig. 3E). Group 4 genes were related to oxidative phosphorylation, non-alcoholic fatty liver disease, and mitochondrial functions. Group 6 genes were associated with the NRF2 pathway and the phagocytosis pathway (Fig. 3F).

Quantitative real-time PCR analysis confirmed that NRF2 target genes, including *UDP-glucose 6-dehydrogenase (Ugdh)*, *glutamate-cysteine ligase catalytic subunit (Gclc)*, *NAD(P)H quinone dehydrogenase 1 (Nqo1)*, and *6-phosphogluconate dehydrogenase (Pgd)*, were significantly upregulated in *Atg7*-KO mouse livers but suppressed in a stepwise manner when wild-type *p62* was replaced with *p62^S351A^*

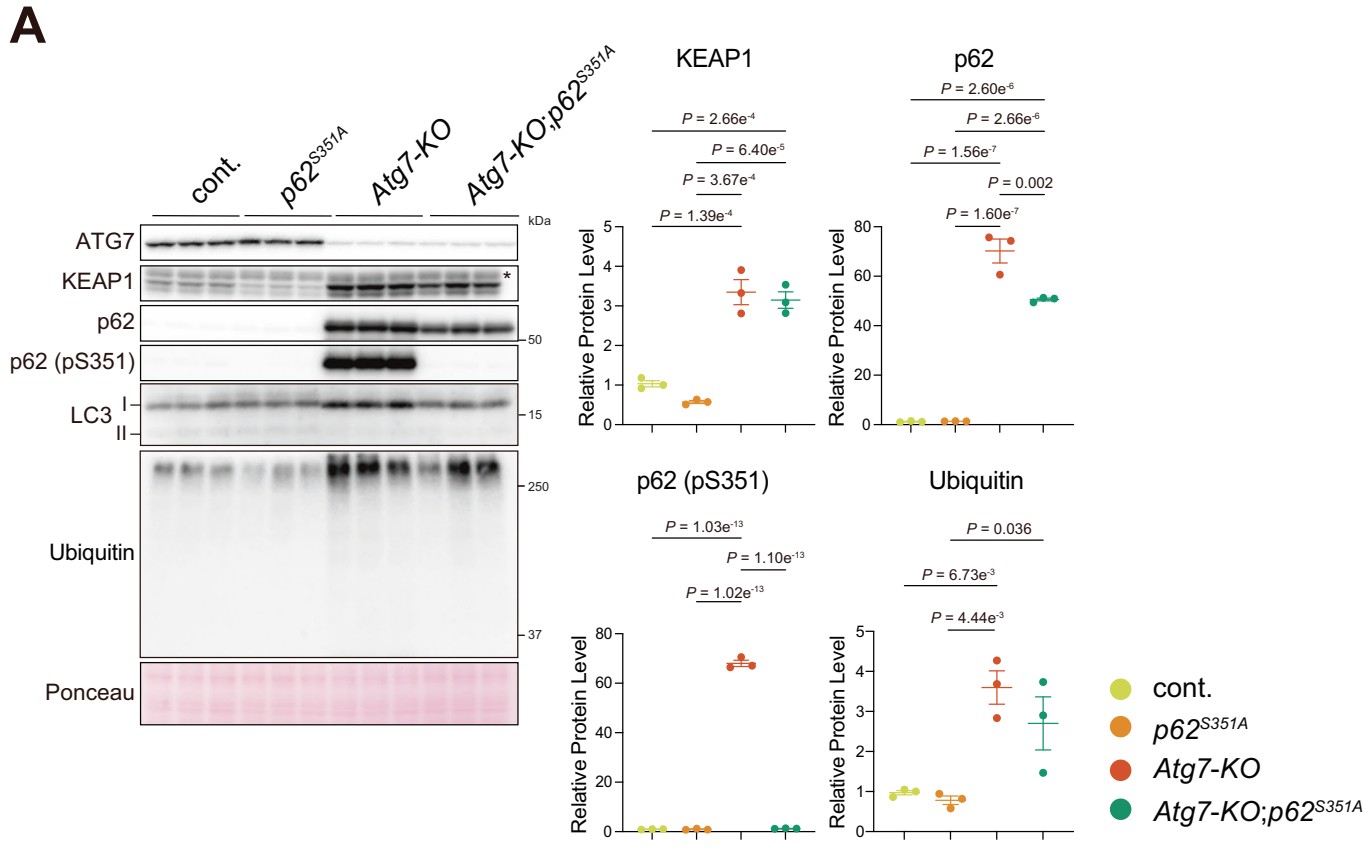

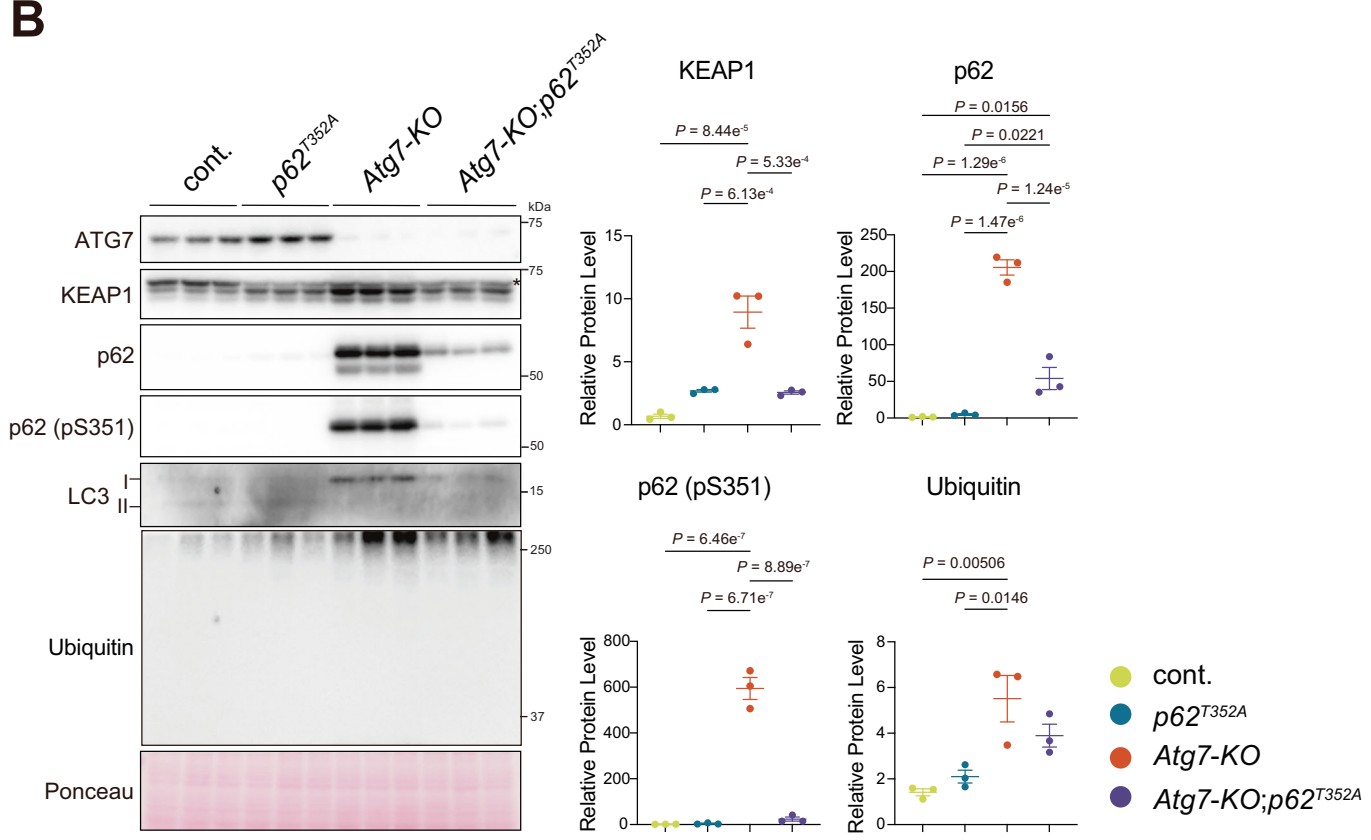

◀ **Figure 2.  KEAP1 levels in *Atg7*-deficient hepatocytes with different *p62* mutations.**

(A, B) Immunoblot analysis. Livers from cont., *p62^{S351A}*, *Atg7*-KO, and *Atg7*-KO;*p62^{S351A}* mice (A) and from cont., *p62^{T352A}*, *Atg7*-KO, and *Atg7*-KO;*p62^{T352A}* mice (B) at 3 month-old were separated into nuclear and cytoplasmic fractions. The cytoplasmic fraction was subjected to SDS-PAGE and analysed by immunoblotting with the indicated antibodies. Cont., in which ATG7 is efficiently expressed at similar level of the wild-type mice were used as control. Graph shows the results of quantitative densitometric analysis of ubiquitinated proteins, KEAP1, p62 and Ser351-phosphorylated p62 relative to the whole protein content estimated using Ponceau-S staining ($n = 3$). Data are means ± s.e. Statistical analysis was performed by Tukey test after one-way ANOVA. Asterisks indicate non-specific bands, as they did not increase in the *Atg7* knockout background. Source data are available online for this figure.

(Fig. 4A) or *p62^{T352A}* (Fig. 4B). Similar to other NRF2 target genes, *Sqstm1* expression was upregulated in *Atg7*-KO livers and reversed by substituting wild-type *p62* with either mutant (Fig. 4B).

Immunoblot analysis showed significant increases in NRF2 target proteins, including NQO1, PGD, GCLC, and GSTM1, in *Atg7*-KO mouse livers. This increase was partially suppressed in *Atg7*-KO;*p62^{S351A}* (Fig. 4C) and completely suppressed in *Atg7*-KO;*p62^{T352A}* livers (Fig. 4D). Nuclear NRF2 levels followed a similar trend, progressively decreasing with reduced KEAP1-p62 binding affinity (Fig. 4C,D). Consistent with these findings, p62 protein levels were significantly elevated in *Atg7*-KO livers compared to cont. livers (Fig. 2A,B), likely due to *Sqstm1* being an NRF2 target gene and the degradation of p62 bodies being impaired by autophagy suppression. In *Atg7*-KO;*p62^{S351A}* livers, p62 protein levels were slightly but significantly lower than in *Atg7*-KO livers. In *Atg7*-KO;*p62^{T352A}* livers, p62 protein levels were dramatically reduced, though still higher than in cont. livers (Fig. 2A,B). These findings align with observations of fewer and smaller p62 bodies in hepatocytes from *Atg7*-KO;*p62^{S351A}* and *Atg7*-KO;*p62^{T352A}* mice compared to *Atg7*-KO mice (Fig. 1A,B). Notably, *Atg7*-KO livers exhibited high levels of Ser351-phosphorylated p62 (Fig. 2A,B), which has high affinity for KEAP1, indicating strong sequestration of KEAP1 within p62 bodies (Fig. 1A,B). Taken together, these results demonstrate that transcriptomic changes caused by *Atg7* deficiency, particularly those involving the NRF2 pathway, are strongly dependent on the binding affinity of KEAP1 to p62 bodies in vivo.

## Proteome-wide changes driven by KEAP1-p62 binding affinity in vivo

Next, we performed cytoplasmic proteome profiling to identify global proteome changes that depend on KEAP1-p62 binding in vivo. Principal component analysis (PCA) of the proteomic data revealed distinct clustering based on genotype (Fig. EV3A), indicating substantial proteomic shifts in *Atg7*-deficient mouse livers compared to cont. livers. Of the approximately 7000 quantified proteins, many showed *Atg7*-dependent changes in abundance (Fig. 5A,B, left panels). As previously reported (Komatsu et al, 2007) and shown in Fig. 4C,D, *Atg7*-deficient mice displayed an accumulation of NRF2 target proteins (Fig. 5A,B, left panels, green dots). Both *Atg7*-KO;*p62^{S351A}* and *Atg7*-KO;*p62^{T352A}* livers partially mitigated this accumulation compared to *Atg7*-KO livers, with *Atg7*-KO;*p62^{T352A}* livers showing a greater reduction in NRF2 target proteins (Fig. 5A,B, right panels), suggesting that this reduction correlates with the inhibition of p62-KEAP1 binding. NRF2 target proteins remained accumulated in *Atg7*-KO;*p62^{S351A}* livers (Fig. 5A, right panel), while their levels were more effectively reduced in *Atg7*-KO;*p62^{T352A}* livers (Figs. 5B, right panel and EV3B). Additionally, the analysis presented in Fig. EV3C indicates that 244

proteins accumulate in both *Atg7*-KO;*p62^{S351A}* and *Atg7*-KO;*p62^{T352A}* livers, compared to cont. livers. These proteins, which accumulate independently of NRF2 activation, represent a group of proteins specifically affected by *Atg7* deficiency. Gene Ontology (GO) analysis of these 244 proteins revealed significant enrichment for components associated with the endoplasmic reticulum (ER), secreted proteins, membranes, microsomes, and lipid droplets (Fig. 5C, left). Further biological process analysis indicated significant enrichment in pathways such as autophagy, phospholipid biosynthesis, phospholipid metabolism, lipid metabolism, pyrimidine biosynthesis, lipid biosynthesis, hemostasis, blood coagulation, and sphingolipid metabolism (Fig. 5C, right). The ER is involved in a wide array of these intracellular metabolic processes, including the synthesis of phospholipids, general lipids, sphingolipids, blood coagulation factors, and even pyrimidines. Moreover, among autophagy-related proteins, we observed an elevation of ER-phagy receptor proteins, such as TEX264, CCPG1, SEC62, and RETREG3 (FAM134C) (Gubas and Dikic, 2022), in the autophagy-deficient background, independent of the p62 mutations (Fig. EV3D). These data underscore the ER as a prominent autophagic target in the *Atg7*-deficient liver. Notably, electron microscopy analysis further confirmed the accumulation of abundant ER structures, including concentric membranous structures originating from the ER (Komatsu et al, 2005), across all autophagy-deficient genotypes examined (*Atg7*-KO, *Atg7*-KO;*p62^{S351A}*, and *Atg7*-KO;*p62^{T352A}* mice) (Fig. EV4), emphasizing that ER-related structures accumulate regardless of NRF2 activation. In summary, these results indicate that, in the autophagy-deficient liver, the predominant alterations involve the accumulation of both NRF2 target proteins and ER-related structures. Although the *p62^{T352A}* mutation effectively reduces the accumulation of NRF2-related proteins, it does not prevent the substantial buildup of ER structures. These findings underscore ER-associated autophagy as a crucial pathway disrupted by the loss of autophagy, aligning with observations from previous studies (Hickey et al, 2023; Zhou et al, 2024).

## Hepatic injury from autophagy suppression is exacerbated by the binding affinity between KEAP1 and p62

Inhibition of autophagy in the mouse liver causes severe hepatomegaly and liver damage, both of which are alleviated by the loss of *Sqstm1* or *Nrf2* (Inami et al, 2011; Komatsu et al, 2010; Ni et al, 2014; Takamura et al, 2011). This suggests that pathogenesis in autophagy-deficient livers is due to KEAP1 sequestration within p62 bodies, leading to continuous NRF2 activation. To investigate this possibility, we examined the phenotypes of *Atg7*-KO, *Atg7*-KO;*p62^{S351A}*, and *Atg7*-KO;*p62^{T352A}*

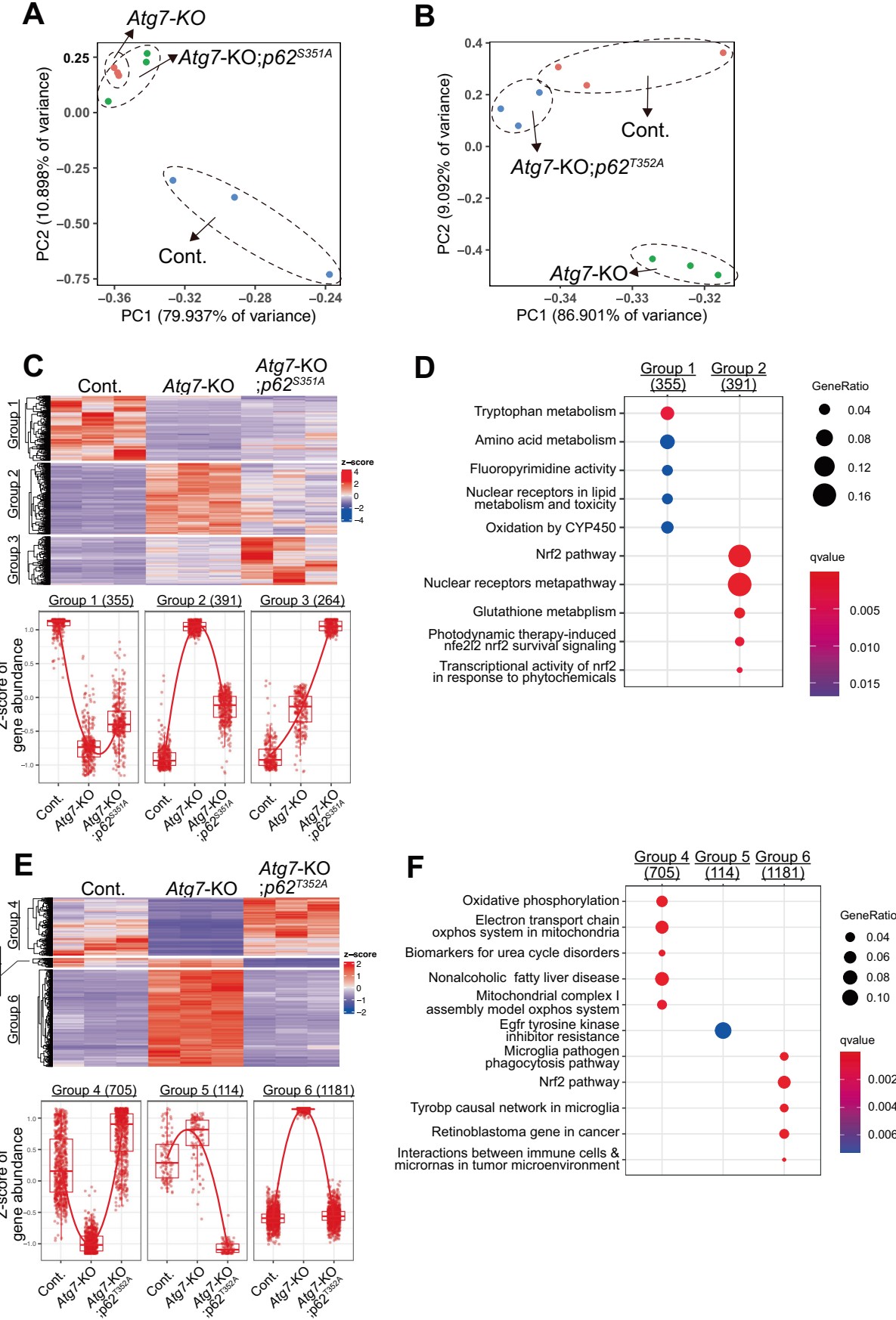

**Figure 3.  Comprehensive transcriptome analyses in *Atg7*-deficient livers with different *p62* mutations.**

(A, B) Principal Component Analysis (PCA). Total RNA was extracted from the livers of cont., *p62*^S3S1A^, *Atg7*-KO, and *Atg7*-KO;*p62*^S3S1A^ mice (A) and from cont., *p62*^T352A^, *Atg7*-KO, and *Atg7*-KO;*p62*^T352A^ mice (B) at 3 months of age. mRNA-seq was performed on three biological replicates per group. (C, D) K-means clustering (C) and pathway analysis (D) of differentially expressed genes (DEGs) from cont., *Atg7*-KO, and *Atg7*-KO;*p62*^S3S1A^ mice. DEGs were identified using DESeq2 (fold change >1.5, FDR < 0.05) and clustered into three groups (C). Heatmaps (upper panels) and boxplots (lower panels) depict the expression patterns of DEGs in each group. The number of DEGs is indicated in parentheses. Pathway enrichment analysis was performed using WikiPathways (D). Box plots show the median (center line), the 25th and 75th percentiles (bounds of the box), and the minimum and maximum values (whiskers). Each dot indicates an individual gene. (E, F) K-means clustering (E) and pathway analysis (F) of DEGs from cont., *Atg7*-KO, and *Atg7*-KO;*p62*^T352A^ mice. DEGs were identified using DESeq2 (fold change >1.5, FDR < 0.05) and clustered into three groups (E). Heatmaps (upper panels) and boxplots (lower panels) depict the expression patterns of DEGs in each group. The number of DEGs is indicated in parentheses. Pathway enrichment analysis was performed using WikiPathways (F). Box plots show the median (center line), the 25th and 75th percentiles (bounds of the box), and the minimum and maximum values (whiskers). Each dot indicates an individual gene. Source data are available online for this figure.

mice. The *p62*^S351A^ and *p62*^T352A^ mice showed no observable phenotype for at least 1 year after birth (data not shown). Consistent with previous studies (Komatsu et al, 2010; Komatsu et al, 2005), *Atg7*-KO mice exhibited significant liver enlargement (Fig. 6A,B). Such hepatomegaly was partially improved by replacing wild-type *p62* with *p62*^S351A^ (Fig. 6A) and completely with *p62*^T352A^ (Fig. 6B). Hematoxylin and eosin staining revealed that hepatocytes in *Atg7*-KO mice were enlarged and filled with eosinophilic materials (Fig. 6C,D). Additionally, acidophilic bodies and pyknotic nuclei were frequently observed (Fig. 6C,D). In *Atg7*-KO;*p62*^T352A^ livers, most hepatocytes appeared normal (Fig. 6D). However, in *Atg7*-KO;*p62*^S351A^ livers, the tissue displayed a mosaic pattern: one population of hepatocytes appeared normal with glycogen-rich areas, while another population exhibited aberrant appearances similar to those observed in *Atg7*-KO mice (Fig. 6C). Although abnormal hepatocytes occupied a large portion (> 90%) of the *Atg7*-KO liver, the affected area was moderately reduced in *Atg7*-KO;*p62*^S351A^ livers and markedly reduced in *Atg7*-KO;*p62*^T352A^ livers (Fig. 6C,D). Since ductular reactions have been observed in autophagy-deficient mouse livers (Barthet et al, 2021; Khambu et al, 2019; Ni et al, 2014), we examined this using CK19 immunostaining. Quantification of the CK19-positive area revealed that the ductular reaction observed in *Atg7*-KO livers was completely suppressed by replacing wild-type *p62* with *p62*^T352A^, but not with *p62*^S351A^ (Fig. 6E,F). Higher serum levels of aspartate aminotransferase (AST), alanine aminotransferase (ALT), and alkaline phosphatase (ALP) were observed in *Atg7*-KO mice, which were partially reduced by substituting wild-type *p62* with *p62*^S351A^ and almost completely with *p62*^T352A^ (Fig. 7A,B). These results suggest that the strength of KEAP1 retention in p62 bodies determines the severity of liver pathology in autophagy-suppressed livers.

## Discussion

p62 is essential for the stress response, but excessive accumulation of p62 is a double-edged sword, as it can cause serious liver enlargement, liver damage, and liver tumorigenesis (Inami et al, 2011; Komatsu et al, 2010; Ni et al, 2014; Takamura et al, 2011). However, the mechanism underlying these liver pathologies remained unknown. In this study, we demonstrated for the first time in vivo that the KEAP1 retention force of p62 bodies is the basis of this mechanism. In other words, we found that NRF2 is activated in proportion to the strength of p62 body's retention force of KEAP1, and that this persistent NRF2 activation in the

combination with autophagy failure causes severe liver enlargement and liver damage (Fig. 7C).

Normally, KEAP1 shuttles between the cytoplasm and p62 bodies (Ikeda et al, 2023). When p62 in the p62 body is phosphorylated at Ser349 by ULK1 kinase or other factors, KEAP1 shifts towards the p62 body side of the equilibrium and is retained in the p62 bodies. As a result, the amount of KEAP1 in the cytoplasm decreases, NRF2 degradation is inhibited, and NRF2 is activated. Ultimately, the p62 bodies containing KEAP1 are broken down by autophagy, and the newly synthesized KEAP1 causes the degradation of NRF2, suppressing NRF2 activation (Komatsu, 2022). How does the NRF2 regulatory mechanism via p62 body cause liver damage? The excessive incorporation of KEAP1 into p62 bodies or the increased binding affinity of p62 for KEAP1 is not sufficient to cause severe liver damage. This is explained by the fact that, in knock-in mice expressing p62^S351E^, in which autophagy is intact but p62 strongly binds to KEAP1, hyperkeratosis is observed in the stratified squamous epithelium of the esophagus and stomach, but the mice showed only mild liver enlargement and damage (Ikeda et al, 2023). This observation suggests that as long as autophagy can rapidly break down the p62 bodies after they incorporate KEAP1, the impact on liver function is likely minimal. Conversely, if autophagy is impaired but KEAP1 incorporation into p62 bodies is limited, the result is only mild liver dysfunction (Figs. 6 and 7). These findings suggest that, in normal cells, the p62 body degradation mechanism coordinates with KEAP1 incorporation into p62 bodies, and that impairment of this coordinated mechanism leads to liver damage.

Notably, p62 is a transcriptional target of NRF2 (Jain et al, 2010). In our in vitro experiments, p62 was overexpressed under a constitutive promoter and thus was not under NRF2 control. The observed difference in p62 body size between in vivo (Fig. 1A,B) and in vitro (Fig. EV2C,D) conditions—namely, the tendency for p62 bodies formed by p62 mutants to be smaller in vivo but not in vitro—may be attributable to differences in the regulatory context of p62 expression. That is, in vivo, p62 expression dynamically changes in response to NRF2 activation, whereas in vitro, p62 expression remains constant. This may explain the observed discrepancy in p62 body size between in vivo (Fig. 1A,B) and in vitro (Fig. EV2C,D) settings.

The regulation of the p62-mediated stress response involves two key rate-limiting steps. The first is the maturation of p62 bodies, driven by LLPS and mediated by interactions between p62 and ubiquitin chains. Phosphorylation of Ser403 and Ser405 within the ubiquitin-binding domain of p62—catalyzed by kinases such as TBK1, CK2, and ULK1—promotes this maturation process (Ikeda

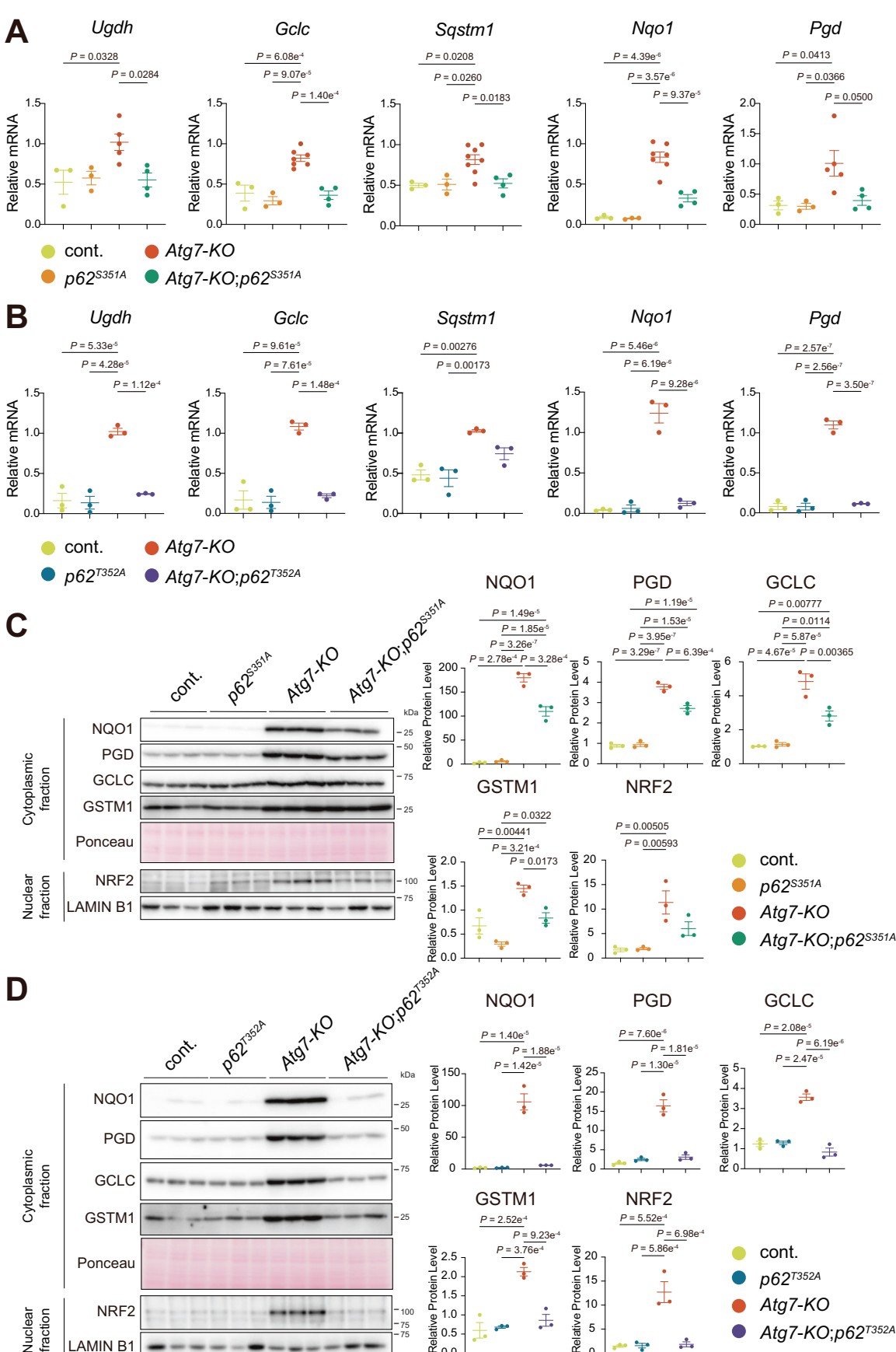

**Figure 4.    NRF2 activation in *Atg7*-deficient livers with different *p62* mutations.**

(A, B) Gene expression of NRF2 targets. Total RNAs were prepared from livers of cont. (*n* = 3), *p62^S351A^* (*n* = 3), *Atg7*-KO (*n* = 5), and *Atg7*-KO;*p62^S351A^* mice (*n* = 4) (A) and of cont. (*n* = 3), *p62^T352A^* (*n* = 3), *Atg7*-KO (*n* = 3), and *Atg7*-KO;*p62^T352A^* mice (*n* = 3) (B) at 3 month-old. Values were normalized against the amount of mRNA in *Atg7*-KO mouse livers. qRT-PCR analyses were performed as technical replicates on each biological sample. Data are means ± s.e. Statistical analysis was performed by Tukey test after one-way ANOVA. (C, D) Immunoblot analysis. Livers from cont., *p62^S351A^*, *Atg7*-KO, and *Atg7*-KO;*p62^S351A^* mice (C) and from cont., *p62^T352A^*, *Atg7*-KO, and *Atg7*-KO;*p62^T352A^* mice (D) at 3 month-old were separated into nuclear and cytoplasmic fractions. Each fraction was subjected to SDS-PAGE and analysed by immunoblotting with the indicated antibodies. Bar graphs show the results of quantitative densitometric analysis of NQO1, PGD, GCLC and GSTM1 relative to the whole protein content estimated using Ponceau-S staining (*n* = 3), and of NRF2 relative to LAMIN B1 (*n* = 3). Data are means ± s.e. Statistical analysis was performed by Tukey test after one-way ANOVA. Source data are available online for this figure.

et al, 2023; Lim et al, 2015; Matsumoto et al, 2011; Pilli et al, 2012). Specifically, TBK1 binds to autophagy receptors NDP52 and TAX1BP1, localizing to p62 bodies via the TBK1 adaptor SINTBAD/NAP1 (Bauer et al, 2024; Yamano et al, 2024). Analogous to Parkin-mediated mitophagy, in which TBK1 clusters on damaged mitochondria by binding directly to Optineurin and initiates TBK1 self-activation essential for mitophagy (Adriaenssens et al, 2024; Nguyen et al, 2023; Yamano et al, 2024), TBK1 may similarly cluster and activate on p62 bodies, a process potentially critical for their maturation and autophagic degradation. This suggests a mechanism in which TBK1 activation is triggered once KEAP1 localizes within p62 bodies. The second critical step is the degradation of p62 bodies, which requires enhanced autophago-some formation on these structures. The ULK1 kinase complex—composed of FIP200, ULK1, ATG101, and ATG13—functions as an upstream factor in autophagosome formation, localizing to p62 bodies where FIP200 directly binds to p62 (Turco et al, 2019). Furthermore, FIP200 can interact with TAX1BP1 and NDP52; the former binds to NBR1, a p62-binding partner, and the latter localizes in p62 bodies (Ravenhill et al, 2019; Vargas et al, 2019). Together, they facilitate additional recruitment of the ULK1 complex to p62 bodies. Because KEAP1 is a major client protein that directly interacts with p62 within p62 bodies, the LLPS of p62 bodies and/or the assembly of the ULK1 complex on p62 bodies may intensify as KEAP1 becomes incorporated. Further research is necessary to elucidate these possible mechanisms in the future.

Why does sustained NRF2 activation and concurrent inhibition of autophagic degradation lead to liver damage? The p62 body serves two primary functions: first, proteostasis through the degradation of ubiquitinated proteins, and second, the activation of NRF2. In *Atg7*-KO;*p62^S351A^* and *Atg7*-KO;*p62^T352A^* mouse livers, the expression of p62 protein is reduced because p62 is one of the targets of NRF2 (Jain et al, 2010). This downregulation suppresses the formation of p62 bodies containing ubiquitinated proteins, potentially increasing soluble ubiquitinated proteins, which may then be degraded through the ubiquitin-proteasome system. Consequently, the reduction of degenerated and then ubiquitinated proteins could help maintain proteostasis, thus alleviating liver damage. However, ubiquitinated proteins still accumulated to similar levels in *Atg7*-KO;*p62^S351A^* and *Atg7*-KO;*p62^T352A^* mice livers, though to a slightly lesser extent than in *Atg7*-KO mice livers (Fig. 2A,B). This suggests that in the absence of autophagy, ubiquitinated substrates that do not form p62 bodies cannot be adequately compensated by the ubiquitin-proteasome system. Therefore, inhibition of ubiquitinated protein degradation is unlikely to be the primary cause of liver damage. Conversely, we found that NRF2 activation, in proportion to the degree of KEAP1

accumulation within p62 bodies, significantly altered the liver transcriptome (Fig. 3) and proteome (Figs. 5 and EV3). Most proteins that accumulated in the *Atg7*-KO mouse livers were NRF2 target proteins, and their levels decreased incrementally in the livers of *Atg7*-KO;*p62^S351A^* and *Atg7*-KO;*p62^T352A^* mice. Proteins commonly accumulating in the livers of *Atg7*-KO;*p62^S351A^*, and *Atg7*-KO;*p62^T352A^* mice included organelle proteins including the ER, along with organelle-specific autophagy receptor proteins such as TEX264, SEC62, and CCPG1, indicating impaired organelle degradation. Among these, excessive accumulation of ER proteins was particularly prominent (Figs. 5, EV3 and EV4). Thus, it is likely that severe liver damage manifests when excessive synthesis of NRF2 target proteins and suppression of organelle degradation—especially of the ER—occur concurrently.

Mallory-Denk Bodies (MDBs) are pathological cytoplasmic structures observed in hepatocytes, and are identified in various liver diseases, including alcoholic liver disease, NASH, and cirrhosis (Stumptner et al, 2002; Zatloukal et al, 2002). Traditionally regarded as inclusions, recent findings from our works suggest that MDBs may exhibit properties of liquid droplets or gel-like structures. These are primarily composed of keratin, ubiquitin, and p62. Studies have shown that MDBs also contain phosphorylated p62 at serine 349 (Ichimura et al, 2013; Saito et al, 2016), KEAP1 (Inami et al, 2011), and other p62 body components, such as Vault (Kurusu et al, 2023), indicating that MDBs may represent either p62 bodies themselves or solidified forms of p62 bodies. Our previous findings suggest that the sequestration of KEAP1 within MDBs leads to sustained NRF2 activation (Ichimura et al, 2013; Inami et al, 2011). The incorporation of KEAP1 into p62 bodies requires phosphorylation of p62 at serine 349 (Ikeda et al, 2023), implying that kinase activation or phosphatase inhibition in liver disorders may be linked to pathogenesis. How are MDBs formed? Impaired degradation of p62 via autophagy has been reported in HCC from NASH in several mouse models (Inokuchi-Shimizu et al, 2014; Nakagawa et al, 2014; Umemura et al, 2016; Zhang et al, 2015), suggesting that defective autophagy is a primary cause. Once KEAP1 is incorporated into p62 bodies, it not only results in chronic NRF2 activation but also interferes with normal autophagic processes. Persistent NRF2 activation elevates *SQSTM1* gene expression (Jain et al, 2010), leading to further formation of p62 bodies. Consequently, selective autophagy becomes unbalanced as it attempts to degrade excessive p62 bodies, resulting in a backlog of organelle degradation and impaired clearance of damaged proteins and organelles—similar to the condition observed in autophagy-deficient mouse livers.

In addition to experimental models, several lines of evidence suggest the clinical relevance of p62 body formation and

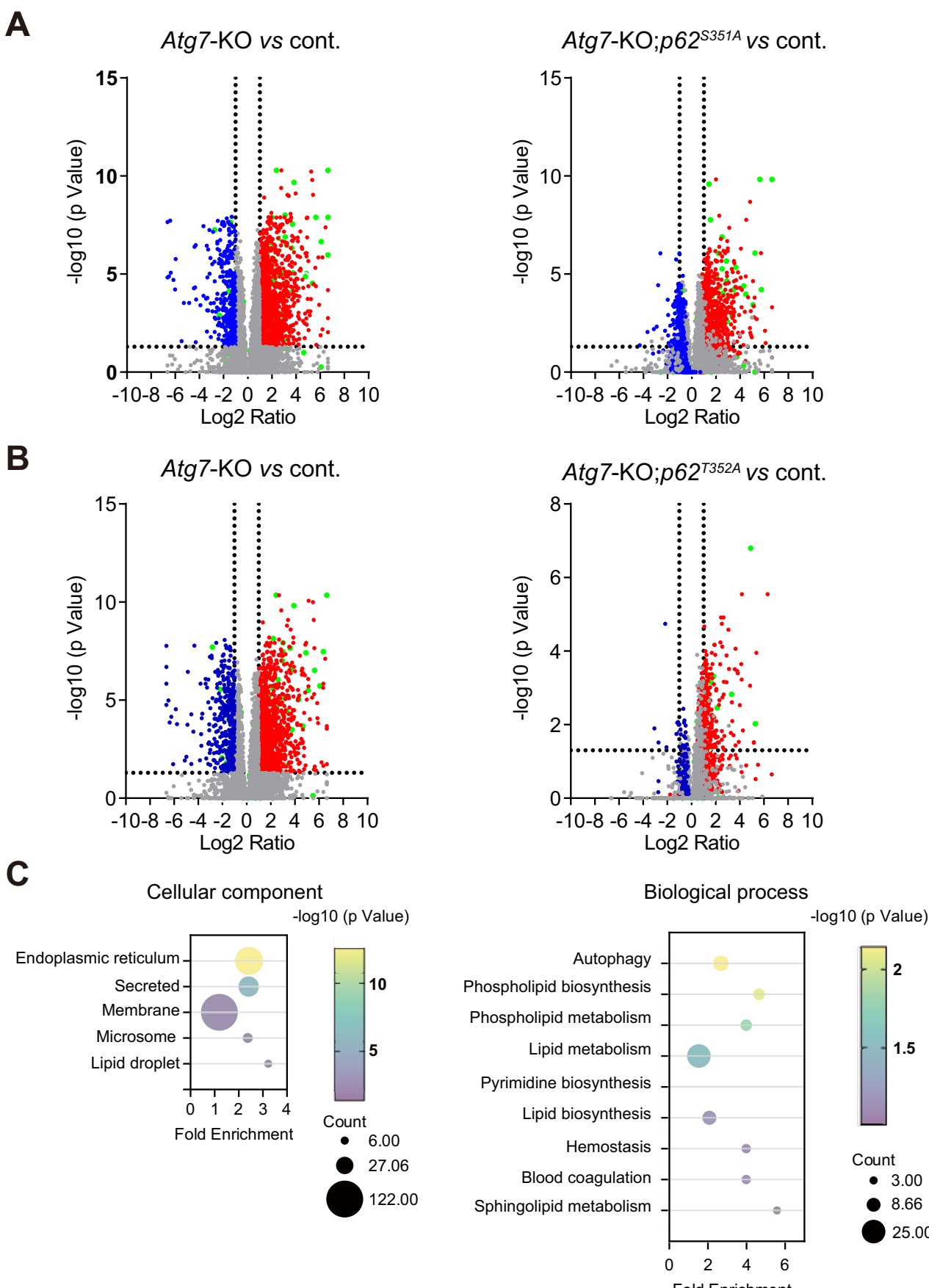

**Figure 5.  Proteome in hepatocyte-specific *Atg7*-knockout mice with different *p62* mutations.**

(A) Proteomic analysis of the liver cytoplasmic fraction from mice. Log2 ratios of *Atg7*-KO (left panel) or *Atg7*-KO;*p62*$^{S351A}$ (right panel) versus cont. are plotted against unadjusted *P* values from one-way ANOVA for individual proteins ($n = 3$). Proteins with a more than 2-fold increase in *Atg7*-KO compared to controls are indicated in red, and those with a more than 2-fold decrease are indicated in blue. NRF2 target genes are marked in green. (B) Proteomic analysis as in (A) was performed for *Atg7*-KO;*p62*$^{T352A}$. (C) Gene Ontology (GO) terms for proteins commonly accumulated in *Atg7*-KO;*p62*$^{S351A}$ and *Atg7*-KO;*p62*$^{T352A}$ mouse livers. The top six GO terms for Cellular Component (left panel) and the top ten GO terms for Biological Process (excluding immunological terms, as inflammation is observed due to Atg7 deficiency) are shown. *P* values were calculated using Fisher's exact test as implemented in DAVID. Source data are available online for this figure.

KEAP1 sequestration in human liver diseases. Cancer-derived KEAP1 mutations have also been reported to influence or promote the formation of p62 bodies (Cloer et al, 2018), though the extent to which p62 phosphorylation, KEAP1 sequestration, and/or autophagy suppression contribute to the underlying pathogenesis needs further investigation. A recent whole-exome sequencing study in individuals with non-alcoholic fatty liver disease (NAFLD) or HCC identified ATG7 loss-of-function variants that impair autophagy and promote p62 accumulation, ballooning, and inflammation (Baselli et al, 2022). These findings underscore the clinical relevance of autophagy suppression and p62 accumulation in liver pathologies. Further investigations using additional in vivo liver disease models—such as NASH and HCC—will be important for defining the broader role of autophagy suppression and KEAP1-p62 interaction in disease progression. Given the genetic analyses in mice in this study, it is conceivable that blocking KEAP1 incorporation into p62 bodies in human liver diseases could serve as a therapeutic target. This could involve ULK1 kinase inhibition, activation of counteracting phosphatases, or disruption of the p62-KEAP1 interaction, all of which may have therapeutic potential as approaches for liver diseases.

# Methods

### Reagents and tools table

| Reagent/resource | Reference or source | Identifier or catalog number |
|---|---|---|
| **Experimental models** | | |
| *p62*-knockout Huh-1 cells | Kurusu et al, 2023 | |
| *p62 FIP200* double knockout Huh-1 cells | This study | |
| *Atg7*$^{flox/flox}$;*Alb-Cre* | Komatsu et al, 2010 | |
| *p62*$^{S351A/S351A}$ knock-in mice | Ikeda et al, 2023 | |
| *p62*$^{T352A/T352A}$ knock-in mice | This study | |
| **Recombinant DNA** | | |
| pMRX-puro p62 retrovirus vector | Komatsu et al, 2010 | |
| pMRX-puro p62$^{S349E}$ retrovirus vector | Ichimura et al, 2013 | |
| pMRX-puro p62$^{S349A}$ retrovirus vector | Ichimura et al, 2013 | |
| pMRX-puro p62$^{T350A}$ retrovirus vector | Komatsu et al, 2010 | |
| pAxCAwtit2 ATG7 adenovirus vector | Saito et al, 2019 | |

| Reagent/resource | Reference or source | Identifier or catalog number |
|---|---|---|
| pAxCAwtit2 ATG7$^{C572S}$ adenovirus vector | Saito et al, 2019 | |
| **Antibodies** | | |
| ATG7 | Cell Signaling Technology | 3936 |
| LC3 | Cell Signaling Technology | 2775 |
| Ubiquitin | Cell Signaling Technology | 3936 |
| KEAP1 | Proteintech Group | 10503-2-AP |
| p62 | MEDICAL & BIOLOGICAL LABORATORIES CO., LTD. | PM066 |
| p62 | BD Biosciences | 610832 |
| p62 | Abnova | H00008878-M01 |
| Ser349-phosphorylated p62 | Ichimura et al, 2013 | |
| NQO1 | Abcam plc | ab34173 |
| PGD | Abcam plc | ab129199 |
| GCLC | Abcam plc | ab41463 |
| UGDH | Abcam plc | ab155005 |
| GSTM1 | Alpha Diagnostic Intl Inc. | GSTM11-S |
| NRF2 | Proteintech Group | 16396-1-AP |
| LAMIN B1 | MEDICAL & BIOLOGICAL LABORATORIES CO., LTD. | PM064 |
| GAPDH | Merck KGaA. Darmstadt | MAB374 |
| CK19 | Proteintech Group | 10712-1-AP |
| Horseradish peroxidase-conjugated Goat Anti-Mouse IgG (H + L) | Jackson ImmunoResearch | 115-035-166 |
| Horseradish peroxidase-conjugated Goat Anti-Rabbit IgG (H + L) | Jackson ImmunoResearch | 111-035-144 |
| Horseradish peroxidase-conjugated Goat Anti-Guinea Pig IgG (H + L) | Jackson ImmunoResearch | 106-035-003 |
| Goat anti-Mouse IgG (H + L) Highly Cross-Adsorbed Secondary Antibody, Alexa Fluor 647 | Thermo Fisher Scientific | A21236 |

| Reagent/resource | Reference or source | Identifier or catalog number |
|---|---|---|
| Goat anti-Rabbit IgG (H + L) Cross-Adsorbed Secondary Antibody, Alexa Fluor 488 | Thermo Fisher Scientific | A11008 |
| Alexa594-Donkey anti-mouse IgG | Jackson ImmunoResearch | 711-545-152 |
| Alexa594-Donkey anti-mouse IgG | Jackson ImmunoResearch | 715-585-151 |
| **Chemicals, enzymes and other reagents** | | |
| Liver Perfusion Kit, Mouse and Rat | Miltenyi Biotec | 130-128-030 |
| Lipofectamine 3000 | Thermo Fisher Scientific | L3000015 |
| cOmplete EDTA-free protease inhibitor cocktail | Roche | 5056489001 |
| bicinchoninic acid (BCA) protein assay | Thermo Fisher Scientific | 23225 |
| NE-PER Nuclear and Cytoplasmic Extraction Reagents | Thermo Fisher Scientific | 78835 |
| gelatin | Sigma-Aldrich | G9391 |
| Hoechst 33342 | Thermo Fisher Scientific | 62249 |
| immunosaver | Nissin EM | 333 |
| FastGene Scriptase Basic cDNA Synthesis | NIPPON Genetics | NE-LS62 |
| TaqMan® Fast Advanced Master Mix | Thermo Fisher Scientific | 444556 |
| TaqMan probe Gapdh | Thermo Fisher Scientific | Mm99999915_g1 |
| TaqMan probe Ugdh | Thermo Fisher Scientific | Mm00447643_m1 |
| TaqMan probe Gclc | Thermo Fisher Scientific | Mm00802655_m1 |
| TaqMan probe Sqstm1 | Thermo Fisher Scientific | Mm0000448091_m1 |
| TaqMan probe Nqo1 | Thermo Fisher Scientific | Mm01253561_m1 |
| TaqMan probe Pgd | Thermo Fisher Scientific | Mm00503037_m1 |
| TRIzol reagent | Thermo Fisher Scientific | 15596-026 |
| NEBNext Poly(A) mRNA Magnetic Isolation Module | New England Biolabs | E7490L |
| NEBNext Ultra II RNA Library Prep Kit for Illumina | New England Biolabs | E7770L |
| 1 M Triethylammonium bicarbonate (TEAB) | Thermo Fisher Scientific | 90114 |
| DTT (Dithiothreitol) | Thermo Fisher Scientific | A39255 |
| Iodoacetamide | Thermo Fisher Scientific | A39271 |
| Phosphoric acid | SIGMA | 438081 |
| S-Trap micro | PROTIFI | C02-micro |

| Reagent/resource | Reference or source | Identifier or catalog number |
|---|---|---|
| Trypsin/Lys-C Protease Mix | Thermo Fisher Scientific | A40009 |
| C18 analytical column | IonOpticks | AUR3-25075C18 |
| **Software** | | |
| Photoshop 2021v25.0 | Adobe | RRID:SCR_014199 |
| FV31S-SW | Olympus | N/A |
| CellPathfinder software | Yokogawa Electric Corp. | N/A |
| Image J software | Schindelin et al, 2012 | RRID: SCR_002285 |
| GraphPad Prism 9 | GraphPad Software | RRID:SCR_002798 |
| RNAseqChef | Etoh and Nakao, 2023 | N/A |
| Proteome Discoverer™ 3.1 | Thermo Fisher Scientific | RRID:SCR_014477 |
| ChatGPT | OpenAI | RRID:SCR_023701 |

## Cell

Huh-1 cells (JCRB0199, NIBIOHN) were cultured in Dulbecco's modified Eagle's medium containing 10% fetal bovine serum, 5 U/mL penicillin, and 50 μg/mL streptomycin. *p62*-knockout Huh-1 cells expressing p62, p62$^{S349E}$, p62$^{S349A}$, or p62$^{T350A}$ were generated using a retroviral vector, as previously reported (Kurusu et al, 2023). Huh-1 cells were authenticated using the STR profile and tested negative for mycoplasma contamination. Primary cultured hepatocytes were isolated from 5-week-old male *Atg7$^{flox/flox}$* and *Atg7$^{flox/flox}$*;Alb-*Cre* mice maintained on a C57BL/6 genetic background using the Liver Perfusion Kit, Mouse and Rat (Miltenyi Biotec, Bergisch Gladbach, Germany) in combination with the gentleMACS™ Dissociator (Miltenyi Biotec).

## Adenovirus Infection

Six-well plates, with or without cover glasses, were collagen-coated before cell seeding. Primary cultured hepatocytes ($2 \times 10^5$ cells per well) were plated and incubated for 12 h. The medium was then replaced with fresh medium containing wild-type ATG7 or ATG7$^{C572S}$ adenovirus (Saito et al, 2019) at a multiplicity of infection (MOI) of 50.

## Mice

*Atg7$^{flox/flox}$*;Alb-*Cre* (Komatsu et al, 2010), and *p62$^{S351A/S351A}$* mice (Ikeda et al, 2023) with the C57BL/6 genetic background were previously described. To generate *p62$^{T352A/T352A}$* knock-in mice using mouse embryonic stem cells (mES cells), we applied the prime editing system as previously described (Ikeda et al, 2023). Briefly, we designed prime-editing guide RNA (pegRNA) containing the following sequences: spacer sequence, 5'-AAAGAAGUGGACCCAUCUAC-3'; reverse transcription template, 5'-GAGUUCACCGGCA-3'; primer-binding site, 5'-GAUGGGUCCAC-3'. The CAG promoter-driven prime editor 2 (PE2) and the U6 promoter-driven pegRNA expression vectors were

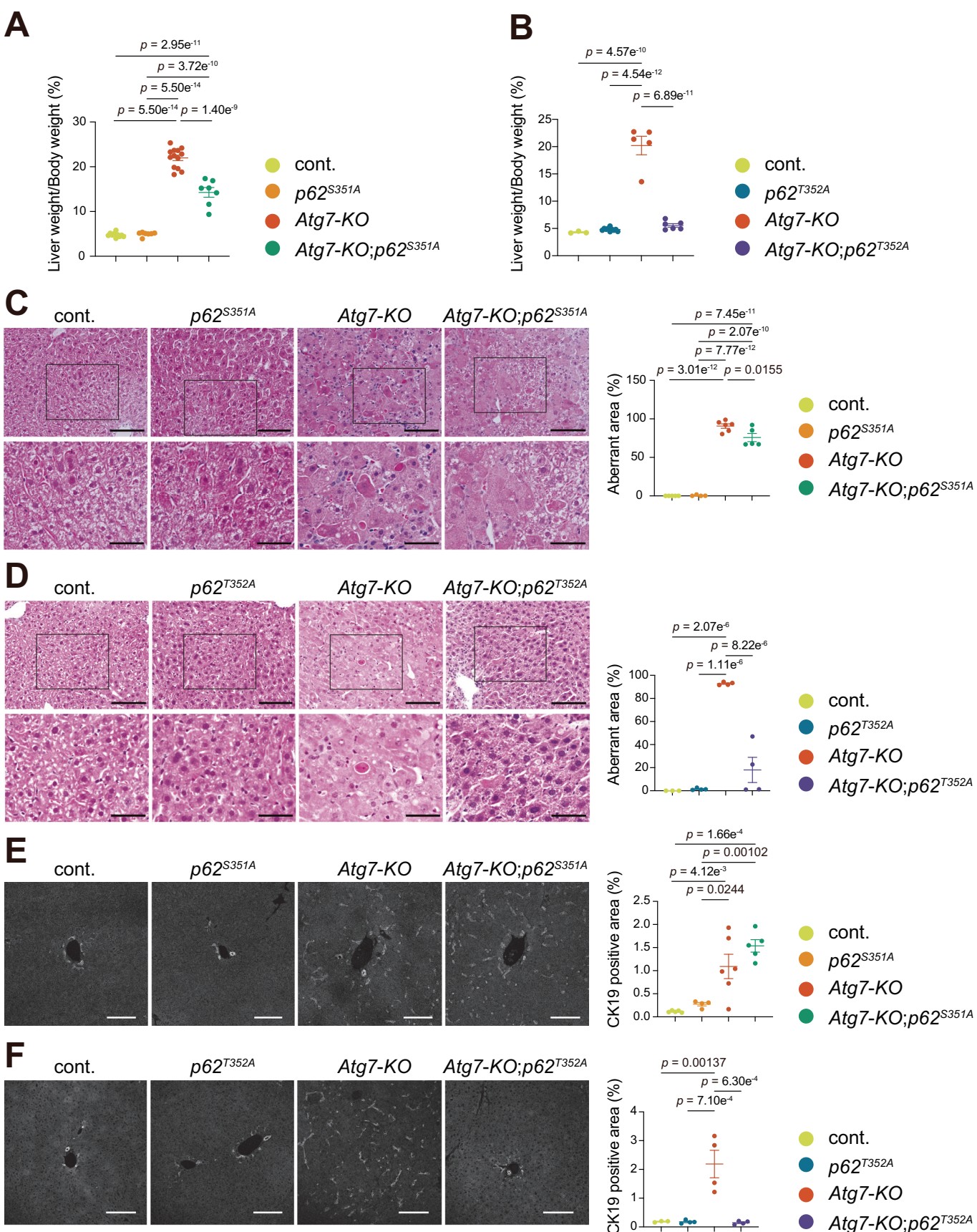

**Figure 6.   Liver pathologies in hepatocyte-specific *Atg7*-knockout mice with different *p62* mutations.**

(A, B) Liver weight (% of body weight) of 3-month-old cont. ($n = 10$), $p62^{S351A}$ ($n = 7$), *Atg7*-KO ($n = 13$), and *Atg7*-KO;$p62^{S351A}$ ($n = 7$) mice (A) and of cont. ($n = 3$), $p62^{T352A}$ ($n = 10$), *Atg7*-KO ($n = 5$), and *Atg7*-KO;$p62^{T352A}$ ($n = 6$) mice (B). Data are means ± s.e. Statistical analysis was performed by Tukey test after one-way ANOVA. (C–F) Hematoxylin and eosin (HE) staining (C, D) and immunohistofluorescence of CK19 (E, F) of livers from cont. ($n = 5$), $p62^{S351A}$ ($n = 4$), *Atg7*-KO ($n = 6$), and *Atg7*-KO;$p62^{S351A}$ ($n = 5$) mice (C, E) and from cont. ($n = 3$), $p62^{T352A}$ ($n = 4$), *Atg7*-KO ($n = 4$), and *Atg7*-KO;$p62^{T352A}$ ($n = 4$) mice (D, F) at 3 month-old. Scale bars, 100 μm (upper rows of (C, F)) and 50 μm (bottom of (C, D)). The area percentage of aberrant hepatocytes (C, D) and CK19-positive regions (E, F) were quantified and shown on the right graphs. Horizontal bars are means ± s.e. Statistical analysis was performed by Tukey test after one-way ANOVA. Source data are available online for this figure.

originally constructed using pCMV-PE2-P2A-GFP (#132776, Addgene) and hU6-sgRNA plasmid. These vectors were co-transfected into RENKA, a C57BL/6N-derived mES cell line, using Lipofectamine 3000 (Thermo Fisher Scientific). Knock-in mutations in transfected mES clones were validated by sequencing of PCR products amplified from genomic DNAs. Culture of mES cells and generation of chimeric mice were carried out as previously described (Mishina and Sakimura, 2007). Biopsies of pup tails were performed for genomic DNA isolation, and mutations were validated by sequencing of PCR products amplified from genomic DNAs. Mice were prepared by mating male and female mice at 2 months of age. All mice were fed *ad libitum* with a standard diet and housed in a specific pathogen-free room maintained at a constant ambient temperature of 21–25 °C, 40–60% of humidity under a 12 h light/dark cycle with free access to food and drink. The Ethics Review Committee for Animal Experimentation of Juntendo University approved the experimental protocol (2022226 and 2022227).

## Immunoblot analysis

Cells were lysed in ice-cold TNE buffer (50 mM Tris-HCl [pH 7.5], 150 mM NaCl, 1 mM EDTA) containing 1% Triton X-100 and cOmplete EDTA-free protease inhibitor cocktail (5056489001, Roche). After centrifugation twice at $15,000 \times g$ for 10 min, the supernatant was collected as the cell lysates. Protein concentrations were determined by bicinchoninic acid (BCA) protein assay (23225, Thermo Fisher Scientific). The lysate was boiled in SDS sample buffer, and the samples were separated by SDS-PAGE and then transferred to polyvinylidene difluoride membranes. Nuclear and cytoplasmic fractions from livers were prepared using the NE-PER Nuclear and Cytoplasmic Extraction Reagents (78835, Thermo Fisher Scientific). SDS-PAGE samples were separated by SDS-PAGE and then transferred to polyvinylidene difluoride (PVDF) membranes. PVDF membranes were stained with Ponceau-S. Antibodies against ATG7 (8558, Cell Signaling Technology, Massachusetts, USA), LC3 (2775, Cell Signaling Technology), Ubiquitin (3936, Cell Signaling Technology), KEAP1 (10503-2-AP, Proteintech Group, Illinois, USA), p62 (PM066, MEDICAL & BIOLOGICAL LABORATORIES CO., LTD., Tokyo, Japan), Ser349-phosphorylated p62 (Ichimura et al, 2013), NQO1 (ab34173, Abcam plc, Cambridge, UK), PGD (ab129199, Abcam plc), GCLC (ab41463, Abcam plc), UGDH (ab155005, Abcam plc), GSTM1 (GSTM11-S, Alpha Diagnostic Intl Inc., TX, USA), NRF2 (16396-1-AP, Proteintech Group), LAMIN B1 (PM064, MEDICAL & BIOLOGICAL LABORATORIES CO., LTD.) and GAPDH (MAB374, Merck KGaA. Darmstadt, Germany) were used as primary antibodies. Blots were then incubated with horseradish peroxidase-conjugated secondary antibody (Goat Anti-Mouse IgG (H + L), 115-035-166, Goat Anti-Rabbit IgG (H + L), 111-035-144, and Goat Anti-Guinea Pig IgG (H + L), 106-035-003, all from Jackson ImmunoResearch, Pennsylvania, USA) and visualized by chemiluminescence.

## Immunofluorescence analysis

Huh-1 cells on coverslips were washed with PBS and fixed with 4% paraformaldehyde (PFA) for 15 min at RT, permeabilized with 0.1% Triton X-100 in PBS for 5 min, and blocked with 0.1% (w/v) gelatin (G9391, Sigma-Aldrich, Darmstadt, Germany) in PBS for 20 min. Then, cells were incubated with primary antibodies in the blocking buffer for 1 h, washed with PBS, and incubated with secondary antibodies for 1 h. Antibodies against p62 (610832, BD Biosciences), and KEAP1 (10503-2-AP, Proteintech Group) were used as primary antibodies. Goat anti-Mouse IgG (H + L) Highly Cross-Adsorbed Secondary Antibody, Alexa Fluor 647 (A21236, Thermo Fisher Scientific) and Goat anti-Rabbit IgG (H + L) Cross-Adsorbed Secondary Antibody, Alexa Fluor 488 (A11008, Thermo Fisher Scientific) were used as secondary antibodies. Nuclei were stained with Hoechst 33342 (62249, Thermo Fisher Scientific). Cells were imaged using the FV3000 confocal laser-scanning microscope with FV31S-SW (Olympus) and a UPlanXApo ×60 NA 1.42 oil objective lens. Contrast and brightness of images were adjusted using Photoshop 2021v25.0 (Adobe, California, USA). The number, and size of p62-positive punctae in each cell and the mean fluorescence intensity of each signal on p62-positive punctae were quantified using a Benchtop High-Content Analysis System (CQ1, Yokogawa Electric Corp., Tokyo, Japan) and CellPathfinder software (Yokogawa Electric Corp.).

## Histological analyses

Mouse livers were excised, cut into small pieces, and fixed by immersion in 4% PFA–4% sucrose in 0.1 M PB, pH 7.4. After rinsing, they were embedded in paraffin for histological analyses. Paraffin sections of 3-μm thickness were prepared and processed for hematoxylin-eosin (HE) staining or immune-histofluorescence (IHF). For IHF, antigen retrieval was performed for 20 min at 98 °C using a microwave processor (MI-77, AZUMAYA, Japan) in 1% immunosaver (Nissin EM, Japan). Sections were blocked and incubated for 2 days at 4 °C with the following primary antibodies: mouse monoclonal antibody against p62 (H00008878-M01, Abnova, Taipei, Taiwan, 1:2000), rabbit polyclonal antibody against KEAP1 (10503-2-AP, Proteintech Group, 1:1000), and/or rabbit polyclonal antibody against CK19 (10712-1-AP, Proteintech Group, 1:1000) followed by Alexa594-Donkey anti-mouse IgG (711-545-152, Jackson ImmunoResearch) and/or Alexa488-Donkey anti-rabbit IgG (715-585-151, Jackson ImmunoResearch). Images of the stained specimens were acquired with a confocal scanning microscope (FV1000, Olympus).

For quantification of HE-stained images, the percentage of areas containing hepatocytes showing hypertrophy and eosinophilic cytoplasm was measured with Image J software (Schindelin et al, 2012). Three different areas per specimen were selected randomly in three to six mice. For CK19-stained images, five areas per

specimen were randomly selected in three to six mice, and the percentage of CK19-positive area was measured with Image J software.

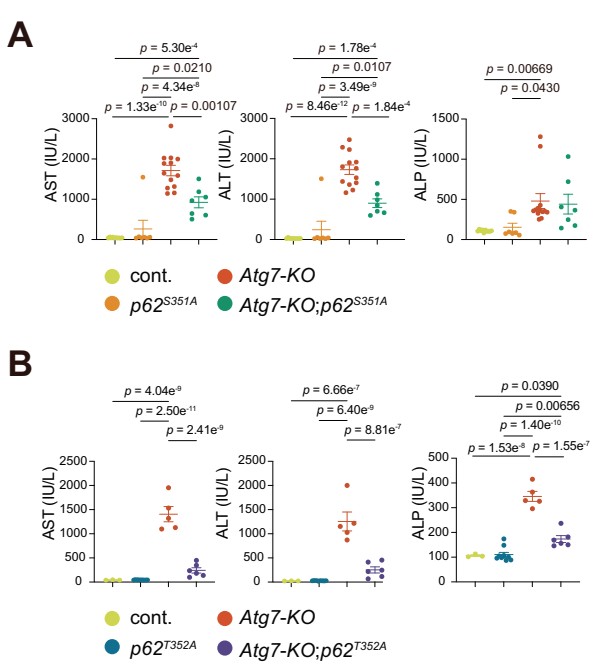

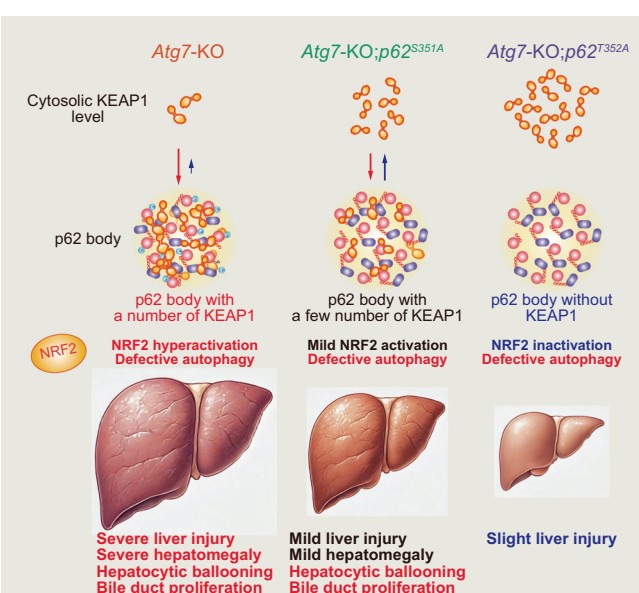

**Figure 7. Liver injury in hepatocyte-specific *Atg7*-knockout mice with different *p62* mutations.**

(A, B) Serum levels of aspartate aminotransferase (AST), alanine aminotransferase (ALT), and alkaline phosphatase (ALP) from cont. ($n = 10$), $p62^{S351A}$ ($n = 7$), *Atg7*-KO ($n = 13$), and *Atg7*-KO;$p62^{S351A}$ ($n = 7$) mice (A) and from cont. ($n = 3$), $p62^{T352A}$ ($n = 10$), *Atg7*-KO ($n = 5$), and *Atg7*-KO;$p62^{T352A}$ ($n = 6$) mice (B) were measured. IU/l, international units/liter. Data are means ± s.e. Statistical analysis was performed by Tukey test after one-way ANOVA. (C) Mechanism of liver injury development induced by autophagy suppression: KEAP1 retention via p62 body. Source data are available online for this figure.

## Electron microscopy

Small pieces of livers were excised and fixed by immersing in 0.1 M PB (pH 7.4) containing 2% PFA and 2% glutaraldehyde. They were post-fixed with 1% osmium tetroxide and 1.5% potassium ferrocyanide in distilled water, embedded in Epon812, and sectioned for observation with an electron microscope (JEM-1400EX; JEOL). Other detailed methods were reported previously (Waguri and Komatsu, 2009).

## Quantitative real-time PCR (qRT-PCR)

cDNAs were synthesized with 1 μg of total RNA using FastGene Scriptase Basic cDNA Synthesis (NE-LS62, NIPPON Genetics, Tokyo, Japan). qRT-PCR was performed with TaqMan® Fast Advanced Master Mix (444556, Thermo Fisher Scientific) on a QuantStudio™ 6 Pro (A43180, Thermo Fisher Scientific). Signals were normalized against *Gapdh* (Glyceraldehyde-3-phosphate dehydrogenase). Predesigned TaqMan Gene Expression Assays, including primer sets and TaqMan probes (Gapdh; Mm99999915_g1, Ugdh; Mm00447643_m1, Gclc; Mm00802655_m1, Sqstm1; Mm0000448091_m1, Nqo1; Mm01253561_m1, Pgd; Mm00503037_m1) were purchased from Thermo Fisher Scientific.

## mRNA Sequencing (mRNA-seq)

Total RNA was extracted from the livers of *Atg7^flox/flox^*, *Atg7^flox/flox^*;Alb-Cre, *Atg7^flox/flox^*;$p62^{S351A/S351A}$, *Atg7^flox/flox^*;Alb-Cre;$p62^{S351A/S351A}$, *Atg7^flox/flox^*;$p62^{T352A/T352A}$ and *Atg7^flox/flox^*;Alb-Cre;$p62^{T352A/T352A}$ mice using the TRIzol reagent (Thermo Fisher Scientific, Waltham, MA, USA). Poly(A) RNA was isolated using the NEBNext Poly(A) mRNA Magnetic Isolation Module (New England Biolabs, Ipswich, MA, USA). cDNA libraries were synthesized using the NEBNext Ultra II RNA Library Prep Kit for Illumina (NEB). Sequencing was performed on a NextSeq 500 sequencer (Illumina, San Diego, CA, USA) with 75-bp single-end reads. The sequencing reads were mapped to the mm10 reference genome using the Spliced Transcripts Alignment to a Reference (STAR) Aligner. Gene expression levels were quantified using RNA-seq by Expectation Maximization (RSEM), and differentially expressed genes (DEGs) among the samples were identified using DESeq2 through the RNAseqChef pipeline (Etoh and Nakao, 2023).

## Preparation of mouse livers for mass spectrometry analysis

Sample preparation for proteomics was performed as previously described (Heunis et al, 2020) with minor modifications. Cytoplasmic fractions suspended in equal volumes of 2× SDS buffer (100 mM TEAB, 10% SDS), and then protein concentrations of the samples were determined by the BCA method. Fifty μg of each fraction was used for proteomics sample preparation by suspension trapping (S-Trap), with minor modifications to that recommended by the supplier (ProtiFi, Huntington, NY, USA). Samples were reduced to 10 mM DTT for 15 min at 37 °C, and subsequently alkylated with 10 mM Iodoacetamide for 15 min at room temperature in the dark. Five μl of 12% phosphoric acid was added to each sample after reduction and alkylation, followed by the addition of 330 μl S-Trap binding buffer (90% methanol in

100 mM TEAB pH 7.5). Acidified samples were added, separately, to S-Trap micro-spin columns and centrifuged at $4000 \times g$ for 1 min. Each S-Trap miclo-spin column was washed with 150 µl S-Trap binding buffer and centrifuged at $4000 \times g$ for 1 min. This process was repeated for three washes. Twenty µl of 50 mM TEAB pH 8.0 containing sequencing-grade Trypsin/LysC mixture (Thermo: 1:10 ratio) was added to each sample, followed by proteolysis for overnight at 37 °C. Peptides were eluted with 50 mM TEAB pH 8.0 and centrifugation at $4000 \times g$ for 1 min. Elution steps were repeated using 0.5% TFA and 0.5% TFA in 50% acetonitrile (ACN), respectively. The three eluates from each sample were combined and dried using a speed-vac before storage at $-80$ °C. Peptides were subsequently reconstituted to 0.1%TFA, 5% ACN, and peptide concentration were calculated by the Quantitative peptide Assay kit (Thermo Fisher Scientific).

### DIA analysis

Each peptide (1.5 µg) was first loaded onto a Vanquish Neo (Thermo Fisher Scientific) connected inline to an Orbitrap Exploris480 (Thermo Fisher Scientific) equipped with a nano electrospray ion source (Thermo Fisher Scientific). Peptides were separated on a C18 analytical column (IonOpticks, Aurora Series Emitter Column, AUR3-25075C18 25 cm × 75 µm, 1.6 µm FSC C18 with a nanoZero fitting) using a 180-min gradient (solvent A, 0.1% FA; and solvent B, 80% ACN/0.1% FA). MS1 data were collected using the Orbitrap (60,000 resolution, 500–860 $m/z$, injection time 100 ms, standard AGC Target). DIA MS2 spectra were collected at $m/z$ 500–860 at 30,000 resolution to set an AGC target of 1000% ($1 \times 10^6$), a maximum injection time of "auto", and HCD collision energies of 30%. The width of the isolation window was set to 6 Da, and overlapping window patterns at $m/z$ 500–860 were used. The data were analyzed using CHIMERYS3.0 intelligent search algorithm (MSAID GmbH) in Thermo Scientific™ Proteome Discoverer™ 3.1. A predicted spectrum library was generated from the mouse fasta database (taxonomy ID = 10090, version 2024-03-27) by INFERYS™ deep learning framework (MSAID GmbH) for all tryptic +1– + 6 peptides between 7 and 30 amino acids in length. The maximum number of missed cleavage sites for trypsin was set to 2. Fixed carbamidomethyl modifications and Oxidation (Met), GlyGly (Lys), and Phosphorylation (Ser, Thr, and Tyr) were selected as variable modifications. Peptide identification was filtered at false discovery rate (FDR) < 0.01.

### MS data processing and visualization

After processing with a Proteome Discoverer 3.1, the exported data were subjected to GO enrichment tests using DAVID (https://david.ncifcrf.gov/; (Huang da et al, 2009; Sherman et al, 2022). Putative NRF2 target gene are selected by GSEA database (https://www.gsea-msigdb.org/gsea/msigdb/human/geneset/NFE2L2.V2.html). Based on the generated data, the volcano plots and bubble plots were visualized using GraphPad Prism 9 (GraphPad).

### Liver function test

Serum aspartate aminotransferase (AST), alanine aminotransferase (ALT), and alkaline phosphatase (ALP) in the mice were measured by SRL (Tokyo, Japan).

### Artificial intelligence (AI) chatbots

ChatGPT (OpenAI) was used to create an illustration of the mouse livers.

### Statistical analysis

Statistical analyses were performed using the unpaired $t$ test (Welch $t$ test), Tukey test after one-way ANOVA. GraphPad Prism 9 (GraphPad Software) was used for the statistical analyses. All tests were two-sided, and $P$ values of <0.05 were considered statistically significant.

## Data availability

The RNA-seq data have been deposited in the NCBI Gene Expression Omnibus (GEO) under accession number GSE292739. The mass spectrometry proteomics data have been deposited to the ProteomeXchange Consortium via the PRIDE partner repository with the dataset identifier PXD061879.

The source data of this paper are collected in the following database record: biostudies:S-SCDT-10_1038-S44319-025-00483-9.

## Peer review information

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

## Acknowledgements

We thank Ms. Tsuguka Kouno for technical assistance. We also thank Yumiko Kurosu and Mutsuko Honda for their help in histological analyses. This work was supported by JSPS KAKENHI Grant Numbers JP19H05706, JP21H004771, 23K20044, 24H00060, 25H01323 (to MK), 24H01901, 23K27134 (to HM), 23KJ1927 (to ST), 23K27351 (to SW), 24K01981 (to HT); AMED Grant Number JP22gm1410004h0003 (to MK), 21gm6410019h0001 (to HM); JST PRESTO Grant Number JPMJPR24NF (to HM); the Takeda Science Foundation (to MK); the Uehara Memorial Foundation (to MK); the Kobayashi Foundation (to MK); and by the Mitsubishi Foundation (to MK). This work was also supported by JSPS KAKENHI Grant Number JP22H04926, Grant-in-Aid for Transformative Research Areas—Platforms for Advanced Technologies and Research Resources "Advanced Bioimaging Support". This study was also supported by the program of the Joint Usage/Research Center for Developmental Medicine, the program of the Research for Inter-University Research Network for High Depth Omics, IMEG, Kumamoto University, and MEXT Promotion of Development of a Joint Usage/Research System Project: Coalition of Universities for Research Excellence Program (CURE) Grant No. JPMXP1323015486. We thank J Sakamaki and K Tabata for critically reading this manuscript.

## Author contributions

**Shuhei Takada**: Data curation; Formal analysis; Investigation; Visualization; Writing—review and editing. **Nozomi Shinomiya**: Data curation; Formal analysis; Validation; Investigation; Visualization; Writing—review and editing. **Gaoxin Mao**: Data curation; Formal analysis; Investigation; Writing—review and editing. **Hikaru Tsuchiya**: Data curation; Formal analysis; Investigation; Visualization; Writing—review and editing. **Tomoaki Koga**: Data curation; Formal analysis; Investigation; Visualization; Writing—review and editing. **Satoko Komatsu-Hirota**: Data curation; Formal analysis; Investigation; Visualization. **Yu-Shin Sou**: Data curation; Formal analysis; Investigation; Visualization. **Manabu Abe**: Resources; Writing—review and editing. **Elena Ryzhii**: Formal analysis; Visualization. **Michitaka Suzuki**: Formal analysis; Visualization. **Mitsuyoshi Nakao**: Supervision. **Satoshi Waguri**: Visualization; Writing—original draft; Writing—review and editing. **Hideaki Morishita**: Supervision; Writing—original draft; Writing—review and editing. **Masaaki Komatsu**: Conceptualization; Resources; Supervision; Funding acquisition; Writing—original draft; Project administration; Writing—review and editing.

Source data underlying figure panels in this paper may have individual authorship assigned. Where available, figure panel/source data authorship is listed in the following database record: biostudies:S-SCDT-10_1038-S44319-025-00483-9.

## Disclosure and competing interests statement

The authors declare no competing interests.

# Expanded View Figures

**Figure EV1. Accumulation of p62 bodies in *Atg7*-deficient hepatocytes.**                                                                                           ▶

(**A**) Immunofluorescence microscopy. Primary cultured hepatocytes isolated from 5-week-old *Atg7$^{flox/flox}$* and *Atg7$^{flox/flox}$*;Alb-*Cre* mice were immunostained with anti-p62 antibody. Scale bars: 10 μm. (**B**) Electron microscopy. Representative electron micrographs of cytoplasmic regions in primary cultured hepatocytes from 5-week-old *Atg7$^{flox/flox}$*;Alb-*Cre* mice. Boxed region is displayed at higher magnification. Typical p62 bodies, along with concentric membranous structures associated with the endoplasmic reticulum (ER), accumulated in *Atg7*-deficient hepatocytes. Arrow indicates the concentric membranous structure, while arrowheads indicate p62 bodies. Scale bars: 5 μm; 500 nm. (**C**) Immunoblot analysis. Wild-type ATG7 or the active-site mutant ATG7$^{C572S}$ was introduced into *Atg7*-knockout primary cultured hepatocytes (isolated from 5-week-old *Atg7$^{flox/flox}$*;Alb-*Cre* mice) using an adenoviral system. Cell lysates were collected at the indicated time points after infection and subjected to immunoblot analysis with the specified antibodies. The bar graphs present the quantitative densitometric analysis of p62, Ser351-phosphorylated p62, and KEAP1, normalized to GAPDH ($n = 3$). Data are presented as means ± s.e. Statistical analysis was performed using a one-way ANOVA followed by Tukey's test. (**D**) Immunofluorescence microscopy. Wild-type ATG7 or the active-site mutant ATG7$^{C572S}$ was introduced into *Atg7*-knockout primary cultured hepatocytes (isolated from 5-week-old *Atg7$^{flox/flox}$*;Alb-*Cre* mice) using an adenoviral system. Immunostaining was performed at the indicated time points after infection with a p62 antibody. The size and number of p62 bodies per cell were quantified ($n = 500$ cells). Horizontal bars indicate medians, boxes represent the interquartile range (25th–75th percentiles), and whiskers extend to 1.5× the interquartile range; individual outliers are shown as points. Statistical analysis was performed using Welch's *t* test. Scale bars: 10 μm (main panels), 1 μm (inset panels). Scale bars: 10 μm. Source data are available online for this figure.

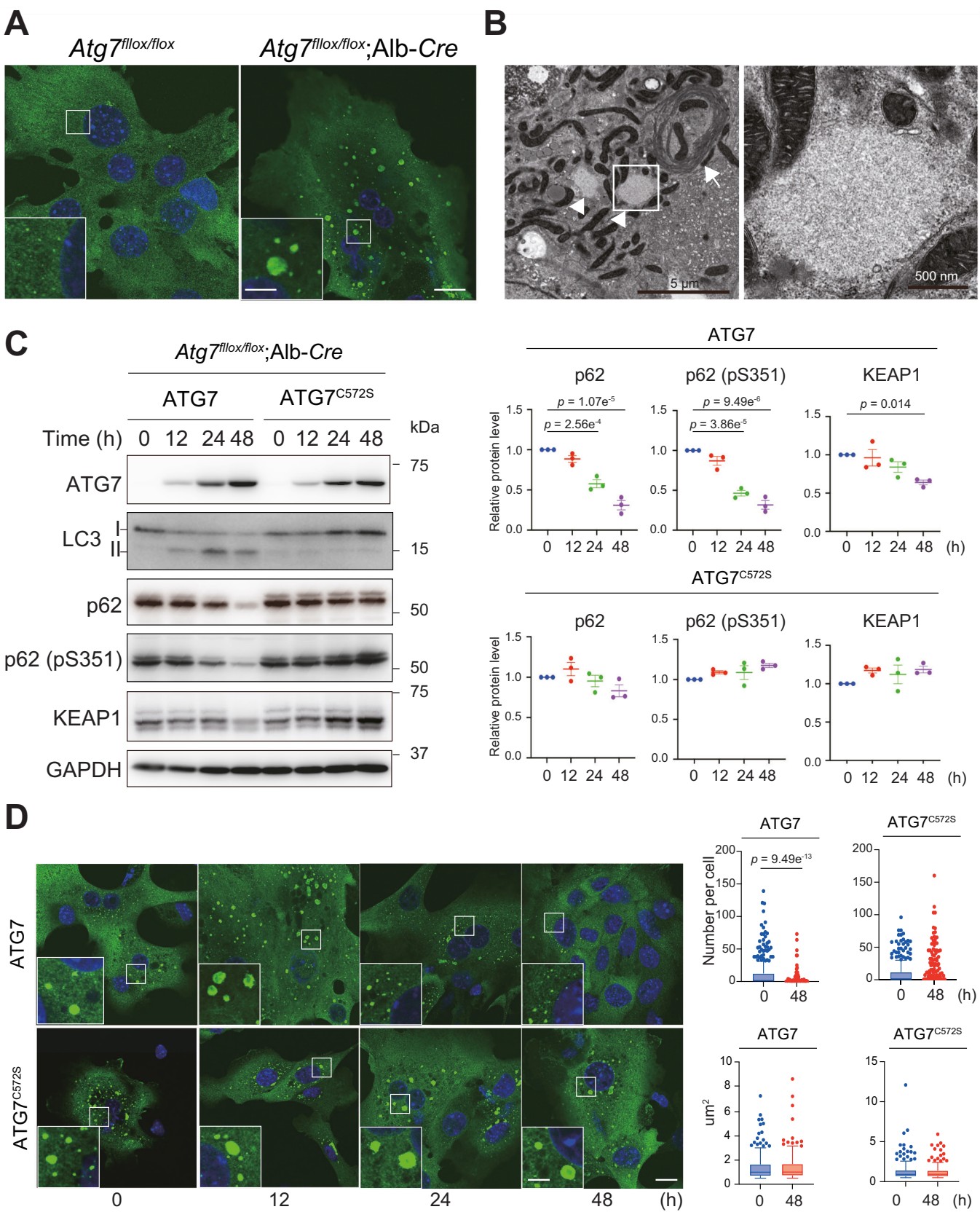

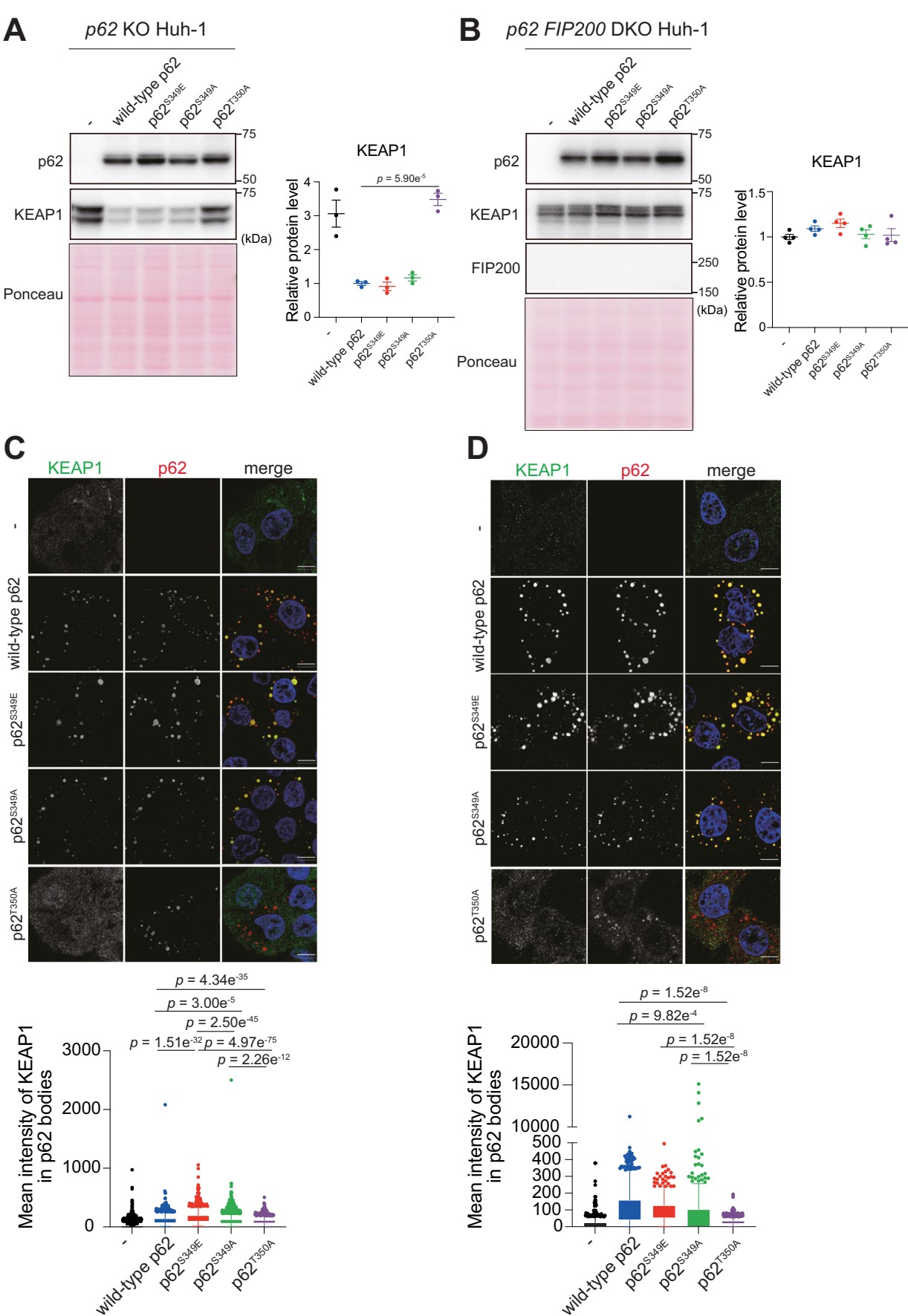

◀ **Figure EV2. KEAP1 protein level in cells expressing p62 mutants.**

(A, B) Immunoblot analysis. Wild-type p62 and indicated p62 mutants were introduced into *p62*-knockout (A) or *p62* and *FIP200*-double knockout (B) Huh-1 cells, and the cell lysates were subjected to immunoblot analysis with indicated antibodies. Bar graph shows the results of quantitative densitometric analysis of KEAP1 relative to the whole protein content estimated using Ponceau-S staining ($n = 3$). Data are means ± s.e. Statistical analysis was performed by Tukey test after one-way ANOVA. (C) Immunofluorescence microscopy. Huh-1 cells indicated in (A) were immunostained with the indicated antibodies. Scale bars, 10 μm (main panels). The graph shows the mean intensity of KEAP1 in p62 bodies comprising of wild-type p62 ($n = 2469$), p62$^{S349E}$ ($n = 1057$), p62$^{S349A}$ ($n = 2992$), or p62$^{T350A}$ ($n = 2033$). Data are means ± s.e. Statistical analysis was performed by Tukey test after one-way ANOVA. (D) Immunofluorescence microscopy. Huh-1 cells indicated in (B) were immunostained with the indicated antibodies. Scale bars, 10 μm (main panels). The graph shows the mean intensity of KEAP1 in p62 bodies comprising of wild-type p62 ($n = 2892$), p62$^{S349E}$ ($n = 2611$), p62$^{S349A}$ ($n = 5191$), or p62$^{T350A}$ ($n = 3522$). Data are means ± s.e. Statistical analysis was performed by Tukey test after one-way ANOVA. Source data are available online for this figure.

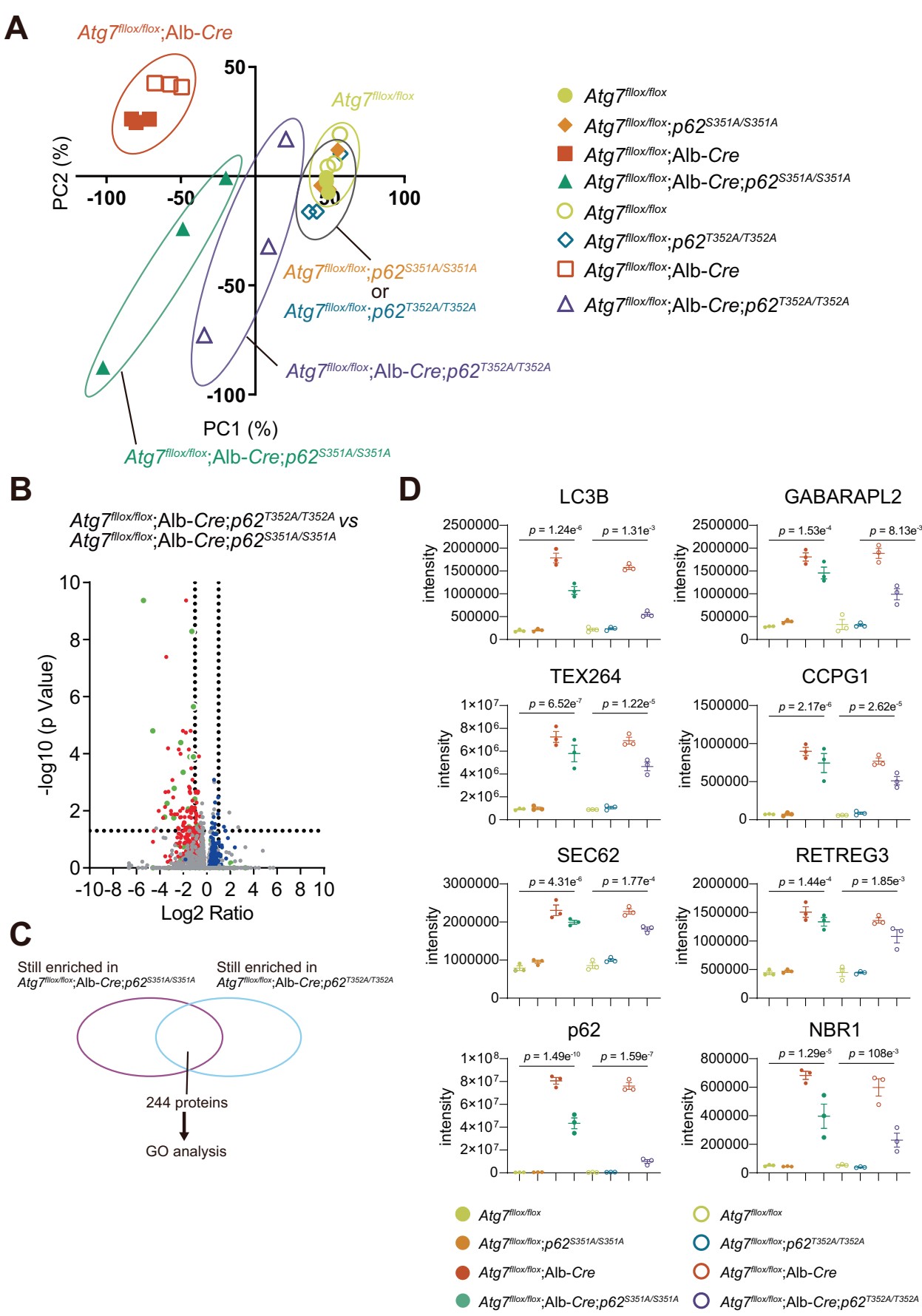

**Figure EV3. Proteomic analysis of hepatocyte-specific *Atg7*-knockout mice with different p62 mutations.**

(A) Principal component analysis (PCA) of mouse liver proteomic data based on triplicate biological replicates. Samples of the same genotype are circled. (B) Proteomic comparison of *Atg7*-KO;*p62*$^{S351A}$ and *Atg7*-KO;*p62*$^{T352A}$. Proteins accumulated as shown in Fig. 5B are indicated in red, and those with decreased abundance are shown in blue. NRF2 target genes are marked in green. (C) Venn diagram of proteomic data showing proteins that accumulated more than 2-fold in both *Atg7*-KO;*p62*$^{S351A}$ and *Atg7*-KO;*p62*$^{T352A}$ mouse livers compared to controls. Gene Ontology (GO) analysis was performed on 244 proteins common to both genotypes. (D) Label-free quantification of autophagy-related and selective autophagy receptor proteins based on MS intensity. The bar graph shows quantitative intensity of autophagy proteins normalized to the total peptide amount, with unadjusted $P$ values from one-way ANOVA for individual proteins ($n = 3$). Data are means ± s.d. Differential protein expressions were assessed using one-way ANOVA, with $P$ values adjusted for multiple comparisons using the Benjamini–Hochberg method. Source data are available online for this figure.

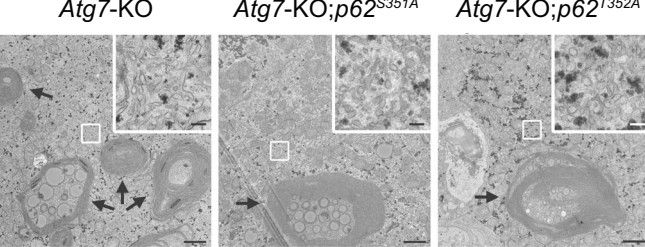

**Figure EV4. Electron micrographs of hepatocytes in hepatocyte-specific *Atg7*-knockout mice with different p62 mutations.**

Electron micrographs of cytoplasmic regions from hepatocytes of the indicated genotypes. Boxed regions are shown at higher magnification in the insets. Concentric membranous structures connected to the endoplasmic reticulum (ER) are present in hepatocytes of all genotypes (arrows). Numerous ER profiles are also visible (insets). Scale bars: 1 µm and 0.2 µm (insets). Source data are available online for this figure.

