## [Peer Review File · EMBO Reports]

KEAP1 retention in phase-separated p62 bodies drives liver damage under autophagy-deficient conditions

Masaaki Komatsu, Shuhei Takada, Nozomi Shinomiya, Gaoxin Mao, Hikaru Tsuchiya, Tomoaki Koga, Satoko Komatsu-Hirota, Yu-Shin Sou, Manabu Abe, Elena Ryzhii, Michitaka Suzuki, Mitsuyoshi Nakao, Satoshi Waguri, and Hideaki Morishita

Corresponding author(s): Masaaki Komatsu (mkomatsu@juntendo.ac.jp)

Review Timeline:

Submission Date:	20th Feb 25
Editorial Decision:	13th Mar 25
Revision Received:	26th Mar 25
Editorial Decision:	22nd Apr 25
Revision Received:	23rd Apr 25
Accepted:	25th Apr 25

Transaction Report:

Dear Masaaki,

Thank you for the submission of your research manuscript to our journal. We have now received the full set of referee reports that is copied below.

As you will see, the referees acknowledge that the findings are interesting and that the conclusions are overall supported by the data presented but they also raise a number of concerns and have suggestions how to further strengthen the data, which should be addressed. Referee 1 and 2 also comment on the somewhat 'artificial' background of ATG7 deficiency and the unclear relevance to liver disease. Though I agree that adding data on p62 bodies and KEAP1 binding in an in vivo liver disease model (or maybe existing material?) would significantly strengthen the pathophysiological relevance of your findings, I understand that setting-up an in vivo liver disease model is beyond the scope of this manuscript. This limitation should however be discussed.

Please let me know in case you disagree, and we can discuss the exact revision requirements further, also in a video chat, if you like.

We realize that it is difficult to revise to a specific deadline. In the interest of protecting the conceptual advance provided by the work, we recommend a revision within 3 months (June 13). Please discuss the revision progress ahead of this time with the editor if you require more time to complete the revisions.

I am also happy to discuss the revision further via e-mail or a video call, if you wish.

*****IMPORTANT NOTE:

We perform an initial quality control of all revised manuscripts before re-review. Your manuscript will FAIL this control and the handling will be delayed IN CASE the following APPLIES:

- 1) A data availability section providing access to data deposited in public databases is missing. If you have not deposited any data, please add a sentence to the data availability section that explains that.
- 2) Your manuscript contains statistics and error bars based on $n=2$. Please use scatter blots in these cases. No statistics should be calculated if $n=2$.

When submitting your revised manuscript, please carefully review the instructions that follow below. Failure to include requested items will delay the evaluation of your revision.*****

- 1) a .docx formatted version of the manuscript text (including legends for main figures, EV figures and tables). Please make sure that the changes are highlighted to be clearly visible.
- 2) individual production quality figure files as .eps, .tif, .jpg (one file per figure). Please download our Figure Preparation Guidelines (figure preparation pdf) from our Author Guidelines pages <https://www.embopress.org/page/journal/14693178/authorguide> for more info on how to prepare your figures.
- 3) a .docx formatted letter INCLUDING the reviewers' reports and your detailed point-by-point responses to their comments. As part of the EMBO Press transparent editorial process, the point-by-point response is part of the Review Process File (RPF), which will be published alongside your paper.
- 4) a complete author checklist, which you can download from our author guidelines (<<https://www.embopress.org/page/journal/14693178/authorguide>>). Please insert information in the checklist that is also reflected in the manuscript. The completed author checklist will also be part of the RPF.
- 5) Please note that all corresponding authors are required to supply an ORCID ID for their name upon submission of a revised manuscript (<<https://orcid.org/>>). Please find instructions on how to link your ORCID ID to your account in our manuscript tracking system in our Author guidelines (<<https://www.embopress.org/page/journal/14693178/authorguide#authorshipguidelines>>)
- 6) We replaced Supplementary Information with Expanded View (EV) Figures and Tables that are collapsible/expandable online. A maximum of 5 EV Figures can be typeset. EV Figures should be cited as 'Figure EV1, Figure EV2' etc... in the text and their respective legends should be included in the main text after the legends of regular figures.

7) Before submitting your revision, primary datasets (and computer code, where appropriate) produced in this study need to be deposited in an appropriate public database (see <<https://www.embopress.org/page/journal/14693178/authorguide#dataavailability>>). Specifically, we would kindly ask you to provide public access to the following datasets:

- mRNA-seq data
- mass spectrometry analysis

The accession numbers and database should be listed in a formal "Data Availability " section (placed after Materials & Method) that follows the model below (see also <<https://www.embopress.org/page/journal/14693178/authorguide#dataavailability>>). Please note that the Data Availability Section is restricted to new primary data that are part of this study.

Data availability

Additional information on source data and instruction on how to label the files are available <<https://www.embopress.org/page/journal/14693178/authorguide#sourcedata>>.

10) Figure legends and data quantification:

- the name of the statistical test used to generate error bars and P values,
- the number (n) of independent experiments (please specify technical or biological replicates) underlying each data point,
- the nature of the bars and error bars (s.d., s.e.m.)
- If the data are obtained from n {less than or equal to} 5, show the individual data points in addition to the SD or SEM.
- If the data are obtained from n {less than or equal to} 2, use scatter blots showing the individual data points.

11) Our journal encourages inclusion of *data citations in the reference list* to directly cite datasets that were re-used and

obtained from public databases. Data citations in the article text are distinct from normal bibliographical citations and should directly link to the database records from which the data can be accessed. In the main text, data citations are formatted as follows: "Data ref: Smith et al, 2001" or "Data ref: NCBI Sequence Read Archive PRJNA342805, 2017". In the Reference list, data citations must be labeled with "[DATASET]". A data reference must provide the database name, accession number/identifiers and a resolvable link to the landing page from which the data can be accessed at the end of the reference. Further instructions are available at <<https://www.embopress.org/page/journal/14693178/authorguide#referencesformat>>.

12) All Materials and Methods need to be described in the main text using our 'Structured Methods' format. According to this format, the Methods section includes a Reagents and Tools Table (listing key reagents, experimental models, software and relevant equipment and including their sources and relevant identifiers) followed by a Methods and Protocols section describing the methods, ideally using a step-by-step protocol format. The aim is to facilitate adoption of the methodologies across labs. Please download and fill our Reagents and Tools Table template (.docx), which you can find in our author guidelines: <https://www.embopress.org/page/journal/14693178/authorguide#structuredmethods>. When submitting your revised manuscript, please do not include the Reagents and Tools Table in the Methods section of the manuscript but upload it as a separate file choosing the file type "Reagent Table". An example of a Method paper with Structured Methods can be found here: <https://www.embopress.org/doi/10.15252/msb.20178071>.

13) As part of the EMBO publication's Transparent Editorial Process, EMBO Reports publishes online a Review Process File to accompany accepted manuscripts. This File will be published in conjunction with your paper and will include the referee reports, your point-by-point response and all pertinent correspondence relating to the manuscript.

Kind regards,

Martina

=====

Referee #1:

KEAP1 targets the key transcription factor for antioxidant response NRF2 for proteasome-mediated degradation. In this paper, Komatsu and colleagues investigated the role of altered KEAP1-p62 interactions in the context of liver pathology. The authors generated p62 knock-in mice with diminished KEAP1 binding (p62S351A/S351A) or lacking KEAP1 retention (p62T352A/T352A) and then crossed these knock-in mice with Atg7flox/flox; Alb-Cre mice, which display severe liver damage, hepatomegaly, and abnormal activation of the Nrf2 pathway mediated by p62 bodies. They found that under autophagy-deficient conditions, p62S351A/S351A and p62T352A/T352A bodies show reduced Keap1 retention and attenuate Nrf2 activation compared to wild-type p62 bodies. Transcriptome and proteome analyses revealed that Nrf2 targets upregulated by Atg7 deficiency are normalized in Keap1-binding deficient p62 mutants. The hepatomegaly and liver damage in Atg7flox/flox; Alb-Cre mice are ameliorated as reduced retention of Keap1 in p62 bodies. These findings provided in vivo evidence to show that Nrf2 activation due to impaired autophagy depends on the binding affinity of Keap1 to p62 bodies.

Major concerns:

1. The size and number of mutant p62 bodies (p62S351A/S351A or p62T352A/T352A) are decreased compared to wild-type p62 in hepatocytes. However, levels of p62 appear not to be affected by mutations in Huh-1 cells (Figure S2A). Does Keap1 regulate formation of p62 bodies?
2. The authors claimed that Keap1 translocation into p62 bodies is required for its autophagic degradation. The level of Keap1

should be compared in p62 KO and ATG7 and p62 double KO cells.

3. How is autophagy impaired in liver diseases with accumulation of p62 bodies? Are p62 bodies in liver diseases similar to p62 bodies in autophagy-deficient cells for Keap1 retention?

4. The Ser349 of p62 is phosphorylated by ULK1, which increases the binding affinity between Keap1 and p62. Does the retention of Keap1 onto p62 bodies show difference in FIP200 KO cells and ATG7 KO cells?

Minor concerns:

1. Page 179, p62S351 should be p62S351A.

2. p62 is constitutively removed by basal autophagy. Is Keap1 degraded by basal autophagy? Are levels of Keap1 as well as Nrf2 targets different in control and p62S351A or p62T352A cells?

Referee #2:

P62-bodies are phase-separated biomolecular condensates that form in certain disease-states and experimentally upon suppression of autophagy. p62 directly binds KEAP1 resulting in displacement of NRF2 and/or KEAP1 recruitment and degradation by the autophagic clearance pathway. p62 bodies indeed show colocalization of KEAP1 and high levels of NRF2-mediated transcription. This manuscript provides important in vivo data to clarify the functional role of the p62-KEAP1 interaction in genetic models of autophagy deficiency. The significance of showing phase separated p62 bodies and their relationship to KEAP1/NRF2 in liver histopathology is high, though somewhat incremental given the teams prior publications which demonstrate in vivo p62-bodies. Human disease relevance is indirect as the manuscript only examines p62-KEAP1/NRF2 in an artificial background of ATG7 deficiency (a weakness). Overall the claims made are well-supported by the data. Given the in vivo nature of the work and the omic-based quantification of organellar changes, the manuscript and its findings will be interesting to many in the liver disease, p62 and KEAP1/NRF2 fields. The primary weakness in the manuscript is that it does not consider KEAP1-p62 binding in p62 bodies of in vivo models of liver disease (eg. NASH, HCC where autophagic clearance of p62 bodies was reported to be disrupted).

Minor comments:

1) The manuscript is grammatically well-written but with some points needing correction. Whether oxidative/electrophilic stress releases NRF2 from KEAP1 is not clear. Likely, CUL3 release is involved for some electrophiles. Cancer-derived mutations in KEAP1 were reported to impact/promote p62-bodies, but with unclear functional relevance to NRF2. For discussion, do the newly presented data inform on NRF2 signaling in KEAP1 mutant cancer or on HCC or on the defective clearance of p62-bodies in disease?

2) A band on the KEAP1 w.blot is noted to be non-specific. Often, KEAP1 runs as multiple bands, some of which can be collapsed with phosphatase. It is likely that the KEAP1 bands on the w.blots in this paper are KEAP1 and NOT non-specific.

3) Quantitative statements are made for data that are not quantified, for example with respect to p62 bodies in number, size, shape etc.

4) If the gain were increased for the IF images, can cytosolic KEAP1 or p62 be visualized? And does this change following ATG7 deletion?

5) The proteomics and RNAseq are clean, strong and confirming to much of what is known. Given that the team has produced both data types, their comparison would be informative to the field.

Referee #3:

The manuscript by Takada et al demonstrates that KEAP1 sequestration by SQSTM1 can exacerbate liver pathology when autophagy is compromised. Previously, this group has shown that deficiency of ATG7 in mouse livers can lead to pathology. When the autophagy receptor Sqstm1 or the transcription factor NRF2 are also knocked out in these mice, the pathology is lessened. One possible model explaining these findings was that accumulation of Sqstm1 led to excessive activation of NRF2 target genes and subsequent pathology. In the present study, the authors tested this model by generating mice with mutations in Sqstm1 that reduced its ability to bind to and sequester the NRF2-inhibitor KEAP1. They found that pathology in liver-specific ATG7 knockout animals correlated with the ability of the mutant Sqstm1 ability to accumulate KEAP1 into phase-separated Sqstm1 bodies. This was also correlated with the expression of NRF2 target genes.

Overall, the study is robust and the authors' conclusions are well supported by the data. As such, I only have a few minor comments.

Figure 2B: Why is the expression of ATG7 increased in ATG7-positive animals expressing the T352A mutant Sqstm1? Is Atg7 a NRF2 target? Is the S352 mutant having a dominant effect? Regardless, the authors should at least comment on the expression of ATG7. Additionally, I think that this figure is miss-labeled: lanes three to six say "p62 S351A", but it should probably say "p62 T352A".

Figure 3C & 3E: Please explain why impacts of Sqstm1 mutations have such different impacts on the expression of "Group 3"

genes. Elsewhere in the manuscript, the S351A mutation had a similar impact to the T352A mutation, resulting in a reversal of the ATG7 knockout phenotype (albeit to a lesser extent). However, the S351A mutation further enhanced the expression of "group 3" genes even beyond what was seen in the ATG7 KO animals. Conversely, the T352A mutant had the opposite effects. Additionally, I found the text describing these results (lines 195-209) to be confusing. In my opinion, this text should be edited for clarity and also to address the question above.

Response to the comments of Reviewer #1

5 *KEAP1 targets the key transcription factor for antioxidant response NRF2 for proteasome-mediated degradation. In this paper, Komatsu and colleagues investigated the role of altered KEAP1-p62 interactions in the context of liver pathology. The authors generated p62 knock-in mice with diminished KEAP1 binding (p62S351A/S351A) or lacking KEAP1 retention (p62T352A/T352A) and then crossed these knock-in mice with Atg7flox/flox; Alb-Cre mice, which display severe liver damage, hepatomegaly, and abnormal activation of the Nrf2 pathway mediated by p62 bodies. They found that under autophagy-deficient conditions, p62S351A/S351A and p62T352A/T352A bodies show reduced Keap1 retention and attenuate*
10 *Nrf2 activation compared to wild-type p62 bodies. Transcriptome and proteome analyses revealed that Nrf2 targets upregulated by Atg7 deficiency are normalized in Keap1-binding deficient p62 mutants. The hepatomegaly and liver damage in Atg7flox/flox; Alb-Cre mice are ameliorated as reduced retention of Keap1 in p62 bodies. These findings provided in vivo evidence to show that Nrf2 activation due to impaired autophagy depends on the binding affinity of Keap1 to p62 bodies.*

Reply

We thank Reviewer #1 for their positive evaluation, valuable comments and insightful suggestions.

Major concerns:

1. *The size and number of mutant p62 bodies (p62S351A/S351A or p62T352A/T352A) are decreased compared to wild-type p62 in hepatocytes. However, levels of p62 appear not to be affected by mutations in Huh-1 cells (Figure S2A). Does Keap1 regulate formation of p62 bodies?*

Reply

We thank this reviewer for the valuable comment. As suggested by the reviewer, we newly assessed the size and number of p62 bodies in p62 KO Huh-1 cells expressing either wild-type or mutant forms of p62. As shown in Figure to Reviewers #1A, when wild-type or mutant p62 was expressed in p62 KO Huh-1 cells, there was no significant difference in the size of p62 bodies. Since this may have been influenced by autophagic degradation, we further examined p62 body size in *FIP200/p62* double KO Huh-1 cells, where autophagic degradation is inhibited, upon expression of wild-type or mutant p62. As shown in Figure to Reviewers #1B, compared to the expression of wild-type p62, p62^{S351A} expression resulted in smaller p62 bodies, whereas p62^{T352A} expression led to larger p62 bodies. This data is inconsistent with the *in vivo* findings and appears to simply depend on expression levels (EV2B). Indeed, the number of p62 bodies also correlated with expression levels (Figure to Reviewers #1C and D).

In these experiments, p62 was overexpressed, rendering its expression level independent of NRF2 regulation. In contrast, under *in vivo* conditions, p62 gene expression is under NRF2 control. Consequently, we believe it is challenging to directly compare the effects of p62 mutations observed in these experiments with *in vivo* findings. This interpretation has been explicitly described in the manuscript (Page 12, lines 331–338 in the revised manuscript).

Regarding KEAP1's potential role in regulating p62 body formation, we fully agree with the reviewer's suggestion. Indeed, our preliminary data indicate that reintroducing a KEAP1 mutant, incapable of binding p62, into *KEAP1* KO Huh-1 cells results in a significant reduction in p62 body size (our unpublished data). To investigate this question more thoroughly, we are currently preparing a separate manuscript focusing on the mechanistic role of KEAP1 in p62 body formation. Therefore, we have deliberately chosen not to include these results in the current manuscript. We greatly appreciate the insightful suggestion and look forward to sharing our findings in the near future.

Figure for referee with unpublished data and its description has been removed upon request by the authors.

55

2. The authors claimed that *Keap1* translocation into p62 bodies is required for its autophagic degradation. The level of *Keap1* should be compared in p62 KO and ATG7 and p62 double KO cells.

60

Reply

We apologize for the confusion caused by our mislabeling. Figures S2A and S2B data were obtained from p62 KO and *FIP200* & p62 double knockout (DKO) cells, respectively. We corrected the typo (Supplementary Figure S2A and B in the revised manuscript). Since *FIP200* functions upstream of *ATG7* (Yamamoto et al., Nat Rev Genet 2023), we consider that the use of *FIP200* & p62 DKO cells provides a more robust suppression of autophagic activity.

65

3. How is autophagy impaired in liver diseases with accumulation of p62 bodies? Are p62 bodies in liver diseases similar to p62 bodies in autophagy-deficient cells for *Keap1* retention?

70

Reply

We thank this reviewer for valuable comment. The mechanism by which accumulation of p62

75

80 bodies suppresses autophagy has been unclear. We speculate that selective autophagy becomes unbalanced as it attempts to degrade excessive p62 bodies, resulting in a backlog of organelle degradation and impaired clearance of damaged proteins and organelles—similar to the condition observed in autophagy-deficient mouse livers (we discussed these points in Discussion section). However, this hypothesis needs further investigation in future.

85 With regard to KEAP1 retention on p62 bodies in liver diseases, our previous work has demonstrated that KEAP1 is sequestered into p62 bodies (also known as Mallory-Denk bodies) in clinical specimens from patients with hepatocellular carcinoma (HCC), which frequently exhibits autophagy suppression (Inami et al., J Cell Biol, 2011; Saito et al., Nat Commun, 2016; Kurusu et al., Dev Cell, 2023).

90 4. The Ser349 of p62 is phosphorylated by ULK1, which increases the binding affinity between Keap1 and p62. Does the retention of Keap1 onto p62 bodies show difference in FIP200 KO cells and ATG7 KO cells?

Reply

95 We have previously reported that ULK1 localizes in p62 bodies through a direct ULK1 - p62 interaction and its localization occurs independently of FIP200 (Ikeda et al., EMBO J 2023). To test this reviewer's suggestion, we investigated the localization of KEAP1 in FIP200 knockout background, and observed that KEAP1 localized to p62 bodies even without FIP200 (EV2D). We provided these new data in EV2D of the revised manuscript and modified the text in the Result section of the revised manuscript (Page 7, lines 184–187 in the revised manuscript).

100

105

Supplementary Fig. S2D Immunofluorescence microscopy. Wild-type p62 and indicated p62 mutants were introduced into p62 and FIP200-double knockout Huh-1 cells, and the cell were immunostained with the indicated antibodies. Scale bars, 10 μ m (main panels). The graph shows the mean intensity of KEAP1 in p62 bodies comprising of wild-type p62 ($n = 2892$), p62^{S349E} ($n = 2611$), p62^{S349A} ($n = 5191$), and p62^{T350A} ($n = 3522$). Data are means \pm s.e. Statistical analysis was performed by Tukey test after one-way ANOVA.

Minor concerns:

1. Page 179, p62S351 should be p62S351A.

110

Reply

We corrected the typo (Page 7, line 178 in the revised manuscript).

115

2. *p62* is constitutively removed by basal autophagy. Is *Keap1* degraded by basal autophagy? Are levels of *Keap1* as well as *Nrf2* targets different in control and *p62*^{S351A} or *p62*^{T352A} cells?

Reply

120

We demonstrated that KEAP1 is degraded in Huh-1 cells in a FIP200-dependent manner (EV2A and B), suggesting that KEAP1 is degraded by basal autophagy. Additionally, KEAP1 levels decreased in *p62* KO Huh-1 cells upon re-expression of wild-type *p62* or the *p62*^{S349A} mutant, but not *p62*^{T350A} mutant (EV2A and B). Regarding the *Nrf2* activity, our previous study showed increased expression levels of *Nrf2* target genes in *p62* KO Huh-1 or 293T cells re-expressing wild-type *p62* or *p62*^{S349A}, whereas *p62*^{T350A} failed to elicit such increase (Figures 1 and 6 in Ichimura et al., Mol Cell 2013). These results collectively indicate that the KEAP1-*p62* interaction drives autophagic degradation of KEAP1 and subsequent *Nrf2* activation.

125

References:

130

Ichimura Y, Waguri S, Sou YS, Kageyama S, Hasegawa J, Ishimura R, Saito T, Yang Y, Kouno T, Fukutomi T et al (2013) Phosphorylation of *p62* activates the Keap1-Nrf2 pathway during selective autophagy. Mol Cell 51(5):618-31.

135

Ikeda R, Noshiro D, Morishita H, Takada S, Kageyama S, Fujioka Y, Funakoshi T, Komatsu-Hirota S, Arai R, Ryzhii E et al (2023) Phosphorylation of phase-separated *p62* bodies by ULK1 activates a redox-independent stress response. EMBO J 42(14):e113349.

Inami Y, Waguri S, Sakamoto A, Kouno T, Nakada K, Hino O, Watanabe S, Ando J, Iwadata M, Yamamoto M et al (2011) Persistent activation of *Nrf2* through *p62* in hepatocellular carcinoma cells. J Cell Biol 193(2):275-84.

140

Kurusu R, Fujimoto Y, Morishita H, Noshiro D, Takada S, Yamano K, Tanaka H, Arai R, Kageyama S, Funakoshi T et al (2023) Integrated proteomics identifies *p62*-dependent selective autophagy of the supramolecular vault complex. Dev Cell 58(13):1189-1205.e11.

Saito T, Ichimura Y, Taguchi K, Suzuki T, Mizushima T, Takagi K, Hirose Y, Nagahashi M, Iso T, Fukutomi T et al (2016) *p62*/*Sqstm1* promotes malignancy of HCV-positive hepatocellular carcinoma through *Nrf2*-dependent metabolic reprogramming. Nat Commun. 7:12030.

145

Yamamoto H, Zhang S, Mizushima N (2023) Autophagy genes in biology and disease. Nat Rev Genet 24(6):382-400

Response to the comments of Reviewer #2

150

P62-bodies are phase-separated biomolecular condensates that form in certain disease-states and experimentally upon suppression of autophagy. *p62* directly binds KEAP1 resulting in displacement of NRF2 and/or KEAP1 recruitment and degradation by the autophagic clearance pathway. *p62* bodies indeed show colocalization of KEAP1 and high levels of NRF2-mediated transcription. This manuscript provides important *in vivo* data to clarify the functional role of the *p62*-KEAP1 interaction in genetic models of autophagy deficiency. The significance of showing phase separated *p62* bodies and their relationship to KEAP1/NRF2 in liver histopathology is high, though somewhat incremental given the teams prior publications which demonstrate *in vivo* *p62*-bodies. Human disease relevance is indirect as the manuscript only examines *p62*-KEAP1/NRF2 in an artificial background of ATG7 deficiency (a weakness).

155

160

Overall the claims made are well-supported by the data. Given the *in vivo* nature of the work

and the omic-based quantification of organellar changes, the manuscript and its findings will be interesting to many in the liver disease, p62 and KEAP1/NRF2 fields. The primary weakness in the manuscript is that it does not consider KEAP1-p62 binding in p62 bodies of *in vivo* models of liver disease (eg. NASH, HCC where autophagic clearance of p62 bodies was reported to be disrupted).

Reply

We sincerely appreciate the reviewer's insightful comments regarding the need for further investigation of *in vivo* models of liver disease, such as NASH and HCC, to explore the broader role of autophagy suppression and KEAP1-p62 interaction in disease progression. We fully agree that such studies would be valuable for a comprehensive understanding of the pathophysiological significance of p62 bodies in liver disease.

Recent whole-exome sequencing studies of NAFLD and HCC patients have identified loss-of-function mutations in ATG7 (P426L and V471A in the catalytic domain), which inhibit autophagy and lead to p62 accumulation, hepatocyte ballooning, and inflammation (Baselli et al., *J Hepatol*, 2022). These findings strongly support the importance of autophagy suppression and p62 accumulation in human liver disease and directly align with the mechanisms investigated in our study. This further underscores the relevance of our work in elucidating the functional consequences of p62-KEAP1 interactions in an ATG7-deficient context. We appreciate the reviewer's suggestion and have incorporated these points into the Discussion section of the revised manuscript (Page 14, lines 406–417).

Minor comments:

1) *The manuscript is grammatically well-written but with some points needing correction. Whether oxidative/electrophilic stress releases NRF2 from KEAP1 is not clear. Likely, CUL3 release is involved for some electrophiles. Cancer-derived mutations in KEAP1 were reported to impact/promote p62-bodies, but with unclear functional relevance to NRF2. For discussion, do the newly presented data inform on NRF2 signaling in KEAP1 mutant cancer or on HCC or on the defective clearance of p62-bodies in disease?*

Reply

We appreciate this valuable and insightful comment. According to the reviewer's suggestion about the Cul3 dissociation model, we have included the following text in the Introduction section of the revised manuscript (Page 4, lines 81–84): When cells experience oxidative stress or electrophilic substances, specific cysteine residues on KEAP1 undergo oxidative modification, leading to the dissociation of KEAP1 from NRF2 or, alternatively, disruption of the KEAP1–CUL3 interaction.

Regarding the crosstalk between the redox- and/or cancer-related KEAP1-dependent and p62-dependent pathways, we consider the following perspective. Since p62 is an NRF2 target (Jain et al., *J Biol Chem* 2010), NRF2 activation leads to p62 accumulation, triggering a positive feedback loop that promotes p62-body formation and NRF2 hyperactivation. In our study, we have demonstrated that KEAP1 sequestration by phosphorylated p62 bodies is an essential step in this feedback loop. A key future challenge is to elucidate the regulatory mechanisms governing p62 phosphorylation and dephosphorylation, and we are currently investigating this process. As noted by this reviewer, cancer-derived KEAP1 mutations have been reported to influence or promote p62-body formation (Cloer et al., *Mol Cell Biol* 2018). However, the extent to which p62 phosphorylation, KEAP1 sequestration, and/or autophagy suppression contribute to the underlying pathogenesis needs further investigation in future. We have added these points to the Discussion section of the revised manuscript (Page 14, lines 406–410).

2) A band on the KEAP1 w.blot is noted to be non-specific. Often, KEAP1 runs as multiple bands, some of which can be collapsed with phosphatase. It is likely that the KEAP1 bands on the w.blots in this paper are KEAP1 and NOT non-specific.

215

Reply

We thank this reviewer for valuable information. The reason we judged the additional band of KEAP1 as non-specific is that it did not increase in *Atg7* knockout background. To avoid any confusion, we have stated this clearly in the legend of Figure 2 of the revised manuscript (Page 25, lines 838–839).

220

3) Quantitative statements are made for data that are not quantified, for example with respect to p62 bodies in number, size, shape etc.

225

Reply

Quantitative evaluation of the number, size, and shape of the p62 bodies shown in Figures 1A and B was technically challenging due to the use of tissue samples. Therefore, we opted to use the phrase “tended to” in the text.

230

4) If the gain were increased for the IF images, can cytosolic KEAP1 or p62 be visualized? And does this change following ATG7 deletion?

Reply

As suggested by this reviewer, we prepared an adjusted version of the Figures 1A and B with increased gain (Figure to Reviewer #2). The reason why p62 bodies are few in wild-type mice compared to autophagy-deficient mice is likely because they are rapidly degraded as soon as they are formed in wild-type mice. The dramatic changes in p62 bodies caused by *Atg7* deficiency are demonstrated in Figures 1A and B.

235

240

Figure to Reviewer #2 Figures 1A and B with increased gain.

5) The proteomics and RNAseq are clean, strong and confirming to much of what is known.

245 Given that the team has produced both data types, their comparison would be informative to the field.

Reply

250 We thank this reviewer for this insightful comment. We fully agree and are currently addressing this issue. However, as this analysis is being conducted as part of a separate study, it is beyond the scope of the present investigation.

References:

255 Baselli GA, Jamialahmadi O, Pelusi S, Ciociola E, Malvestiti F, Saracino M, Santoro L, Cherubini A, Dongiovanni P, Maggioni M et al (2022) Rare ATG7 genetic variants predispose patients to severe fatty liver disease. *J Hepatol* 77(3):596-606.

Cloer EW, Siesser PF, Cousins EM, Goldfarb D, Mowrey DD, Harrison JS, Weir SJ, Dokholyan NV, Major MB (2018) p62-Dependent Phase Separation of Patient-Derived KEAP1 Mutations and NRF2. *Mol Cell Biol* 38(22):e00644-17.

260 Inami Y, Waguri S, Sakamoto A, Kouno T, Nakada K, Hino O, Watanabe S, Ando J, Iwadate M, Yamamoto M et al (2011) Persistent activation of Nrf2 through p62 in hepatocellular carcinoma cells. *J Cell Biol* 193(2):275-84.

265 Jain A, Lamark T, Sjøttem E, Larsen KB, Awuh JA, Overvatn A, McMahon M, Hayes JD, Johansen T (2010) p62/SQSTM1 is a target gene for transcription factor NRF2 and creates a positive feedback loop by inducing antioxidant response element-driven gene transcription. *J Biol Chem* 285: 22576-22591

Kurusu R, Fujimoto Y, Morishita H, Noshiro D, Takada S, Yamano K, Tanaka H, Arai R, Kageyama S, Funakoshi T et al (2023) Integrated proteomics identifies p62-dependent selective autophagy of the supramolecular vault complex. *Dev Cell* 58(13):1189-1205.e11.

270 Saito T, Ichimura Y, Taguchi K, Suzuki T, Mizushima T, Takagi K, Hirose Y, Nagahashi M, Iso T, Fukutomi T et al (2016) p62/Sqstm1 promotes malignancy of HCV-positive hepatocellular carcinoma through Nrf2-dependent metabolic reprogramming. *Nat Commun.* 7:12030.

275 **Response to the comments of Reviewer #3**

The manuscript by Takada et al demonstrates that KEAP1 sequestration by SQSTM1 can exacerbate liver pathology when autophagy is compromised. Previously, this group has shown that deficiency of ATG7 in mouse livers can lead to pathology. When the autophagy receptor Sqstm1 or the transcription factor NRF2 are also knocked out in these mice, the pathology is lessened. One possible model explaining these findings was that accumulation of Sqstm1 led to excessive activation of NRF2 target genes and subsequent pathology. In the present study, the authors tested this model by generating mice with mutations in Sqstm1 that reduced its ability to bind to and sequester the NRF2-inhibitor KEAP1. They found that pathology in liver-specific ATG7 knockout animals correlated with the ability of the mutant Sqstm1 ability to accumulate KEAP1 into phase-separated Sqstm1 bodies. This was also correlated with the expression of NRF2 target genes.

Overall, the study is robust and the authors' conclusions are well supported by the data. As such, I only have a few minor comments.

290 Reply

We thank Reviewer #3 for the positive evaluation of our study and for the thoughtful comments.

1. Figure 2B: Why is the expression of ATG7 increased in ATG7-positive animals expressing the T352A mutant Sqstm1? Is Atg7 a NRF2 target? Is the S352 mutant having a dominant effect?

295 *Regardless, the authors should at least comment on the expression of ATG7. Additionally, I think that this figure is miss-labeled: lanes three to six say "p62 S351A", but it should probably say "p62 T352A".*

Reply

300 We thank this reviewer for this important comment. As suggested by this reviewer, we have quantified the level of ATG7 in Figure 2B, and found that there was no significant difference between cont. and p62^{T352A} (Figure to Reviewer #3). We corrected the typo in Figure 2B in the revised manuscript.

305 **Figure to Reviewer #3** Quantification of ATG7 levels in Figure 2B in the revised manuscript. Graph shows the results of quantitative densitometric analysis of ATG7 relative to the whole protein content estimated using Ponceau-S staining ($n = 3$). Data are means \pm s.e. Statistical analysis was performed by Tukey test after one-way ANOVA.

310 2. Figure 3C & 3E: Please explain why impacts of *Sqstm1* mutations have such different impacts on the expression of "Group 3" genes. Elsewhere in the manuscript, the S351A mutation had a similar impact to the T352A mutation, resulting in a reversal of the ATG7 knockout phenotype (albeit to a lesser extent). However, the S351A mutation further enhanced the expression of "group 3" genes even beyond what was seen in the ATG7 KO animals. Conversely, the T352A mutant had the opposite effects. Additionally, I found the text describing these results (lines 195-209) to be confusing. In my opinion, this text should be edited for clarity and also address the question above.

Reply

320 We sincerely appreciate this reviewer's valuable comment. We apologize for any confusion in the legends in the panels and the text descriptions. The genes categorized as "Group 3" in Fig. 3C and those in Fig. 3E are entirely different. To ensure clarity and avoid any misunderstanding, we have revised the legends in the revised manuscript to explicitly distinguish these groups as follows:

325 Fig. 3C, D: Group 1–3
Fig. 3E, F: Group 4–6

Dear Masaaki,

Thank you for the submission of your revised manuscript to EMBO reports and for your patience while it was evaluated by former referee #1. As you can see, the referee considers all concerns adequately addressed during the revision and recommends publication.

Before I can accept the manuscript, I need you to address some minor points below:

- Please update the 'Conflict of interest' paragraph to our new 'Disclosure and competing interests statement'. For more information see

<https://www.embopress.org/page/journal/14693178/authorguide#conflictsofinterest>

- Regarding the Author Contributions, we now use CRediT to specify the contributions of each author in the journal submission system. Therefore, please remove the Author Contributions from the manuscript file and make sure that the author contributions in our online manuscript tracking system are correct and up-to-date. The information you specified in the system will be automatically retrieved and typeset into the article. You can enter additional information in the free text box provided, if you wish.

- We encourage formal citation of preprints in the reference list, in your case to Berkamp et al, 2024. The citation in the text is: (preprint: NAME1 et al, YEAR); in the reference list: Author NAME1, Author NAME2 (YEAR) article title. bioRxiv doi [PREPRINT].

- I noticed that the reference Bauer et al, 2024 in the reference list lacks information on the issue.

- The information on funding in the Acknowledgments and in the online manuscript tracking system must match. In this respect, we noticed the following points that need your attention:

1) Discrepancy between JP22gm1410004h0003 in the manuscript vs. JP22gm1410004h000 in the system.

2) Missing in the online system: the Kobayashi Foundation; Grant-in-Aid for Transformative Research Areas — Platforms for Advanced Technologies and Research Resources "Advanced Bioimaging Support"; Joint Usage/Research Center for Developmental Medicine, the program of the Research for Inter-University Research Network for High Depth Omics, IMEG, Kumamoto University; MEXT Promotion of Development of a Joint Usage/Research System Project: Coalition of Universities for Research Excellence Program (CURE) Grant No. JPMXP1323015486.

- Materials and methods should be Methods

- Data availability section: Please add specific URLs that resolve directly to the datasets PXD061879, GSE292739. Moreover, this section is meant to refer exclusively to data deposited in public repositories. Therefore please remove the following statement: "Additional datasets used and/or analyzed during this study are available from the corresponding author upon reasonable request."

- Our production/data editors have asked you to clarify several points in the figure legends (see below). Please incorporate these changes in the manuscript and return the revised file with tracked changes with your final manuscript submission.

A) Figure legend text:

- Please note that the legend for figure 3 is not provided in a sequential manner. This needs to be rectified.

B) Statistical test information.

- Please indicate the statistical test used for data analysis in the legend of figure 5C

C) Replicates and error bars:

- Please note that the box plots need to be defined in terms of minima, maxima, centre, bounds of box and whiskers, and percentile in the legends of figures 3C, E.

- Please note that information related to n is missing in the legends of figures 4A, B.

D) Data presentation:

- Please note that the error bars are not defined in the legend of figure EV3 D

- Please sort the source data. We need one folder per figure with subfolders for the panels. E.g., the folder for Figure 1 would contain a subfolder for panel A, that contains the images and the quantification. The subfolder B contains the images shown in B and the quantification as .xls file etc.

- As a standard procedure, we edit the title and abstract of manuscripts to make them more accessible to a general readership. Please find the suggested versions below my signature.

- Finally, EMBO Reports papers are accompanied online by

A) a short (1-2 sentences) summary of the findings and their significance,

B) 2-3 bullet points highlighting key results and

C) a schematic summary figure that provides a sketch of the major findings (not a data image).

Please provide the summary figure as a separate file in PNG or JPG format at a size of 550x300-600 pixels (width x height).

Please note that the size is rather small and that text needs to be readable at the final size. Please send us this information along with the revised manuscript.

With kind regards,

Martina

=====

Referee #1:

My concerns have been addressed.

=====

KEAP1 retention in phase-separated p62 bodies drives liver damage under autophagy-deficient conditions

Phase-separated p62 bodies activate NRF2, a key transcription factor for antioxidant response, by sequestering KEAP1, which targets NRF2 for degradation. Although p62 bodies containing KEAP1 are degraded by autophagy, they accumulate in various liver disorders. Their precise disease role remains unclear. We show that excessive KEAP1 retention in p62 bodies and NRF2 activation are major causes of liver damage when autophagy is impaired. In mice with weakened or blocked p62-KEAP1 interactions, KEAP1 retention and NRF2 activation under autophagy-deficient conditions were suppressed. Transcriptome and proteome analyses reveal that p62 mutants unable to bind KEAP1 normalize the expression of NRF2 targets induced by defective autophagy. Autophagy deficiency causes organelle accumulation, especially of the ER, regardless of p62 mutation. Liver damage and hepatomegaly resulting from autophagy suppression markedly improved in mice carrying p62 mutants, particularly those with blocked KEAP1 binding. These findings highlight excessive KEAP1 retention in p62 bodies and defective organelle turnover as key drivers of liver pathology, underscoring the significance of phase separation in vivo.

The authors have addressed all minor editorial requests.

Dr. Masaaki Komatsu
Juntendo University School of Medicine
Department of Physiology
Hongo 2-1-1
Bunkyo-ku, Tokyo 113-8421
Japan

Dear Masaaki,

Thank you very much for implementing the last minor edits. I am very pleased to accept your manuscript for publication in the next available issue of EMBO reports. Thank you for your contribution to our journal.

Kind regards,

Martina
